# SAMPLE-EFFICIENT MULTICLASS CALIBRATION UNDER $\ell_p$ ERROR

## ABSTRACT

Calibrating a multiclass predictor, that outputs a distribution over labels, is particularly challenging due to the exponential number of possible prediction values. In this work, we propose a new definition of calibration error that interpolates between two established calibration error notions, one with known exponential sample complexity and one with polynomial sample complexity for calibrating a given predictor. Our algorithm can calibrate any given predictor for the entire range of interpolation, except for one endpoint, using only a polynomial number of samples. At the other endpoint, we achieve nearly optimal dependence on the error parameter, improving upon previous work. A key technical contribution is a novel application of adaptive data analysis with high adaptivity but only logarithmic overhead in the sample complexity.

## 1 INTRODUCTION

Trustworthiness and interpretability have become key concerns for machine learning models, especially as they are increasingly used for critical decision making. Calibration is an important tool, dating back to classical forecasting literature (Dawid, 1982; Foster & Vohra, 1998), that can be used to address some of these concerns. A predictor $h$ for binary classification that outputs values in $[0, 1]$ is calibrated if, among the inputs $x$ for which $h(x) = q$, exactly $q$ fraction of them have a positive outcome. In recent years, a large body of work has focused on developing algorithms that either learn calibrated predictors or calibrate previously trained models. This notion has also been extended to multi-calibration (Hébert-Johnson et al., 2018), where the calibration guarantee holds for multiple, possibly overlapping populations. Another important extension is to the multiclass setting, which is the focus of this work.

Calibration presents two main challenges. The first is defining a notion of calibration error that quantifies how much a predictor deviates from being perfectly calibrated. This error metric must be testable (Rossellini et al., 2025), meaning that we should be able to detect that a predictor has small error using a small number of samples. While sharing common intuition, many different definitions of calibration error exist in the literature. Typically, the predicted probabilities are divided into bins and the calibration guarantee applies to conditioning on the bins rather than on the predicted values. However, some proposed error metrics are not testable. For example, the $L_\infty$ error as defined by Gruber & Buettner (2022) measures the maximum conditional deviation between the prediction and the true probability of the class across bins. This maximum could occur in a bin containing points that appear with very small probability, making it practically undetectable due to insufficient sampling. The second challenge is developing algorithms that efficiently learn a calibrated predictor from scratch or recalibrate existing predictors, considering both the sample complexity and the computational efficiency with respect to the problem parameters.

While discretizing the prediction space results in a reasonable number of bins for binary classification, in the multiclass setting the number of bins grows exponentially with the number of classes, presenting a unique challenge. In fact, for a natural definition of the distance to calibration, testing whether a given model is perfectly calibrated requires the number of samples to be exponential in the number of classes (Gopalan et al., 2024). A consequence of this result is that estimating a commonly used calibration error metric, one that generalizes the binary classification case to multiclass classification by summing the errors over all the classes and bins, requires a number of samples exponential in the number of classes Gopalan et al. (2024). Alternatively, the

works of Haghtalab et al. (2023) and Dwork et al. (2023) have considered a weaker definition where the predictor is considered calibrated if the calibration error per bin is small, as opposed to measuring the total error across all bins. In this case, surprisingly, a calibrated predictor can be found using a polynomial number of samples. A natural question is whether the weakening in the definition is necessary and, if so, how much weakening is necessary to remove the exponential dependence on the number of classes.

Calibration is important in its own right, but it is also desirable for a predictor to be accurate. Given that most machine learning models are developed using complex pipelines that are difficult to modify, the ability to calibrate an existing model, as opposed to building a new one from scratch, is valuable. This approach would allow one to leverage the remarkable accuracy of existing models while adding calibration guarantees. Moreover, it is possible for a predictor to be calibrated yet uninformative. This underscores the importance of maintaining accuracy alongside calibration. While many works in the literature satisfy this requirement, the works with strong sample complexity bounds of Haghtalab et al. (2023) and Dwork et al. (2023) unfortunately do not. Thus, a significant challenge is to develop efficient algorithms that can calibrate a given predictor while making minimal targeted modifications. Concretely, we aim to develop calibration algorithms for a given predictor that satisfy the following two properties:

1. The resulting classifier is calibrated up to error $\varepsilon$.

2. The resulting classifier's accuracy remains within an additive error of $\varepsilon$ compared to the accuracy of the given predictor to allow for discretization and estimation error.

In this work, we address the above questions and propose a new definition of calibration error, which we call the $\ell_p$ calibration error. This error notion is defined as the $\ell_p$ norm of the calibration errors across all bins and classes. In particular, for a fixed bin and class, we define the calibration error as the product of the absolute difference between the expected value of the prediction and the true probability of the class conditioned on the datapoint belonging to the bin, and the probability mass of the bin. The definition that adds up the errors across all bins and classes corresponds to the special case $p = 1$, also known as the expected calibration error (ECE) (Dawid, 1982), while the definition that measures the maximum error across all bins and classes in Haghtalab et al. (2023) corresponds to $p = \infty$. As our measure of accuracy, we use the squared error of the predictor. Any algorithm that calibrates a predictor to achieve small $\ell_1$ calibration error (ECE) requires exponentially many samples in $k$ Gopalan et al. (2024). Our work shows that for all $p > 1$, there exists an algorithm that uses a polynomial number of samples in the number of classes to calibrate any given predictor. For the special case $p = \infty$ and a given desired calibration error $\varepsilon$, the sample complexity is within a poly-logarithmic factor of $O\left(1/\varepsilon^2\right)$. This is almost as good as one could hope for since even testing if the fraction of data with positive outcome is $1/2$ or $1/2 + \varepsilon$ already requires $\Omega\left(1/\varepsilon^2\right)$ samples.

**Theorem 1** (Informal version of Theorem 7). *There exists an algorithm that takes as input any $k$-class predictor $f : \mathcal{X} \to \Delta_k$, runs in time polynomial in $k$ and $\frac{1}{\varepsilon}$, and, using $\tilde{O}\left(\left(\frac{2^{1/(p-1)}}{\varepsilon^{p/(p-1)}}\right)^2\right)$ samples, returns a $k$-class predictor $h : \mathcal{X} \to \Delta_k$ that has:*

1. *$\ell_p$ calibration error at most $\varepsilon$, and*

2. *squared error within an additive term $\tilde{O}\left(\frac{\varepsilon^{p/(p-1)}}{2^{1/(p-1)}}\right)$ from the squared error of $f$.*

The $\tilde{O}$ notation hides logarithmic factors in $k$ and $1/\varepsilon$.

## 1.1 OUR TECHNIQUES

When $p = \infty$, we observe that if a bin contains at most an $\varepsilon$ fraction of the data distribution, its calibration error for any class is also bounded by $\varepsilon$. Thus, one only needs to care about $1/\varepsilon$ bins with large probability masses. We generalize this idea to all $\ell_p$ norms for $p > 1$ and allow the algorithm to focus only on bins with large probability masses. This observation is sufficient to obtain a (large) polynomial sample complexity. This approach works because our calibration error notion incorporates the probability mass of the bin in the $p$-exponent, naturally assigning higher weights to larger bins.

A second observation that further improves the sample complexity is that for interpretability reasons the outpus of our calibrated predictor should be probability distributions over the $k$ labels, a constraint not enforced in previous work. This constraint significantly reduces the discretized prediction space during calibration compared to $\lambda^k$ in prior works (where $\lambda$ is the number of discrete values per coordinate), since the predictor outputs must form valid probability distributions with coordinates summing to 1. Consequently, our set of bins approximately corresponds to the set of sparse vectors in $k$ dimensions containing $\lambda$ non-zero elements, each equal to $1/\lambda$. The crucial insight is that the number of such sparse vectors is polynomial rather than exponential in $k$.

Calibrating the predictor might require adaptively merging many high-probability bins together. Naively estimating the error of all subsets of high-probability bins to $\varepsilon$ requires $1/\varepsilon^3$ samples (due to the number of subsets being $\Omega\left(\exp\left(1/\varepsilon\right)\right)$). Adaptive data analysis has been applied in previous works to reduce the number of samples, but the overhead remains polynomial in $1/\varepsilon$. Surprisingly, our algorithm is still highly adaptive, but with a novel analysis, the overhead in the sample complexity is only logarithmic in $1/\varepsilon$. Our techniques might be applicable to other problems where adaptive data analysis is used.

## 1.2 RELATED WORK

The most closely related works are by Haghtalab et al. (2023) and Dwork et al. (2023). In the case where $p = \infty$, they showed that with access to an oracle for the exact probabilities, $O\left(\varepsilon^{-2}\ln k\right)$ oracle queries suffice to find an $\varepsilon$-calibrated predictor for $k$-class classification. These results construct a new model from scratch and do not aim to preserve the accuracy of a previously trained model, as our algorithm does. Furthermore, Haghtalab et al. (2023) showed that $O\left(\ln(k)/\varepsilon^4\left(\ln(1/(\varepsilon)) + \ln(V)\right)\right)$ samples suffice for their algorithm, where $V$ is the number of discretized bins. In their case, $\ln(V) = O\left(k\ln(\lambda)\right)$, with $\lambda$ being a non-negative integer that controls the granularity of discretization. In contrast, our algorithm employs a different discretization scheme where $\ln(V) = O\left(\min\left(k, \lambda\right)\ln\left(\lambda + k\right)\right)$. This alternative approach contributes to our improved sample complexity. However, it introduces additional complexity to the algorithm due to the need to project the predictions onto the probability simplex. These projections impact both the calibration and the accuracy of the predictor. For calibration, updating one coordinate of the predictor and then projecting can alter other coordinates that are already calibrated. For accuracy, we must carefully select the projection method that we use to ensure that the accuracy is preserved.

Many calibration algorithms are iterative and, thus, inherently present an adaptive data analysis challenge, due to the dependence of the bins whose predictions get updated on the current predictor. Most algorithms in this area, including ours, perform $\text{poly}\left(1/\varepsilon\right)$ iterations. Some works, such as Gopalan et al. (2022), address the adaptivity issue by resampling at each iteration to estimate the calibration error, which results in a $\text{poly}\left(1/\varepsilon\right)$ overhead in sample complexity. Other works use tools from adaptive data analysis to bound the sample complexity in a black-box way (Haghtalab et al., 2023; Hébert-Johnson et al., 2018). Specifically, they use the strong composition property of differential privacy, which allows answering $t$ adaptive queries with only a $\tilde{O}\left(\sqrt{t}\right)$ overhead. As a result, this method incurs a smaller $\text{poly}(1/\varepsilon)$ overhead in sample complexity. Our novel algorithm and analysis achieve a tighter bound, requiring only a $\log(1/\varepsilon)$ overhead in sample complexity. This significantly improves the overall sample complexity of the iterative calibration process.

Due to the challenges of calibration in the multiclass setting, several weaker error definitions have been proposed. A lot of work focuses on calibrating existing neural networks. For instance, Guo et al. (2017) introduced confidence calibration, where the conditioning is done on the highest prediction value among all classes, and explored several methods including binning methods, matrix and vector scaling, and temperature scaling. Related notions include top-label calibration (Gupta & Ramdas, 2022), which conditions on the highest prediction value and the identity of the top class, and class-wise calibration (Kull et al., 2019), which conditions on individual class predictions rather than on the entire probability vector. While extensive literature exists on $\ell_p$-style calibration measures (Kumar et al., 2019; Vaicenavicius et al., 2019; Widmann et al., 2019; Zhang et al., 2020; Gruber & Buettner, 2022; Popordanoska et al., 2022), our approach differs fundamentally. We incorporate the probability mass of the bin in the $p$-exponent, ensuring that bins with large error have also sufficient mass for detection, resolving the limitation that previously considered $\ell_p$ calibration errors may require exponentially many samples for testing. On the theoretical front, Gopalan et al.

(2022) proposed low-degree multi-calibration as a less-expensive alternative to the full requirement and Gopalan et al. (2024) introduced projected smooth calibration as a multiclass calibration error definition for efficient algorithms with strong guarantees.

## 2 PRELIMINARIES

We use $\mathcal{X}$ to denote the feature space and $[k] = \{1, \ldots, k\}$ to denote the label space. We also use the $k$-dimensional one-hot encoding of a label as an equivalent representation. We use $\Delta_k$ to denote the probability simplex over $k$ labels. In this work, we consider that a $k$-class predictor $f$ is a function that maps feature vectors in $\mathcal{X}$ to distributions in $\Delta_k$.

Instead of conditioning on the exact predicted probability vector, we partition $\Delta_k$ into level sets. Previous methods partition $\Delta_k$ by mapping the prediction vectors to the closest vector in $L^k$, the $k$-ary Cartesian power of $L = \{0, 1/\lambda, 2/\lambda, \ldots, 1\}$, where $\lambda$ is a positive integer that determines the discretization granularity. Note that the coordinates of vectors in $L^k$ may not sum to 1. We use an alternative partition of $\Delta_k$ via a many-to-one mapping onto $V_\lambda^k$. We define $V_\lambda^k$ to be the subset of $L^k$ such that for every member $v$ of $V_\lambda^k$, there exists a probability distribution $u \in \Delta_k$ such that $v$ is obtained by rounding down every coordinate of $u$ to a multiple of $1/\lambda$. Formally,

$$V_\lambda^k = \left\{ v \in L^k : \exists u \in \Delta_k \text{ s.t. } \lfloor u_i \lambda \rfloor / \lambda = v_i \; \forall i \in [k] \right\}.$$

**Example 2.** For $k = 3$ classes and $\lambda = 2$ the set of vectors $V_\lambda^k$ is

$$V_2^3 = \{(0, 0, 0), (0.5, 0, 0), (0, 0.5, 0), (0, 0, 0.5), (0, 0, 1),$$
$$(0, 1, 0), (1, 0, 0), (0, 0.5, 0.5), (0.5, 0, 0.5), (0.5, 0.5, 0)\}.$$

While vectors in $V_\lambda^k$ are not necessarily distributions, they are close to vectors that are distributions. This property allows $V_\lambda^k$ to be significantly smaller than $L^k$.

**Lemma 3.** *For any $\lambda, k \in \mathbb{N}^+$, the number of level sets in $V_\lambda^k$ is at most $\binom{\lambda+k}{k}$. Note that $\log\left(|V_\lambda^k|\right) = O\left(\min(k, \lambda) \ln(\lambda + k)\right)$ whereas $\log\left(|L^k|\right) = O\left(k \ln(\lambda)\right)$.*

The proof of Lemma 3 is provided in the Appendix.

We define the rounding function $R : \Delta_k \to V_\lambda^k$, which maps a prediction vector to the corresponding level set in $V_\lambda^k$: $R(u)_i = \lfloor u_i \lambda \rfloor / \lambda \; \forall i \in [k]$. Conversely, we define the function $\rho$ that maps a level set $v \in V_\lambda^k$ to the closest canonical distribution $\rho(v) = \arg\min_{u \in \Delta_k, R(u)=v} \|u - v\|_\infty$. Finally, we define the projection function $\pi : [0, 1]^k \to \Delta_k$ in $\ell_2$ norm : $\pi(v) = \arg\min_{u \in \Delta_k} \|u - v\|_2$. In some cases, we abuse notation by writing $f(S)$ to denote the common value of a function $f(x)$ for all $x \in S$, when $f(x) = f(y)$ for all $x, y \in S$.

For our sample complexity results, we use the following lemmas for adaptive data analysis and concentration of measure.

**Lemma 4.** *(Jung et al., 2020, Theorem 23) Let $A$ be an algorithm that, having access to a dataset $S = \{x_i\}_{i \in [n]}$, interactively takes as input a stream of queries $q_1, \ldots, q_t : \mathcal{X} \to [0, 1]$ and provides a stream of answers $a_1, \ldots, a_t \in [0, 1]$. Suppose that $A$ is $(\varepsilon, 0)$-differentially private and that*

$$\mathbb{P}\left[ \max_{j \in [t]} \left| \frac{1}{n} \sum_{i \in [n]} q_j(x_i) - a_j \right| \geq \alpha \right] \leq \beta.$$

*Then, for any $\eta > 0$, $\mathbb{P}\left[ \max_{j \in [t]} |\mathbb{E}_{x \sim P}[q_j(x)] - a_j| \geq \alpha + e^\varepsilon - 1 + \sqrt{\frac{2 \ln(2/\eta)}{n}} \right] \leq \beta + \eta$.*

**Lemma 5.** *(Chung & Lu, 2006, Theorem 3.6) Suppose $X_1, \ldots, X_n$ are independent random variables with $X_i \leq M$ for all $i$. Let $X = \sum_{i=1}^n X_i$ and $\|X\| = \sqrt{\sum_{i=1}^n \mathbb{E}[X_i^2]}$. Then,*

$$\mathbb{P}\left[X \geq \mathbb{E}[X] + \lambda\right] \leq \exp\left(-\frac{\lambda^2}{2\left(\|X\|^2 + M\lambda/3\right)}\right).$$

## 3 MULTICLASS CALIBRATION UNDER $\ell_p$ ERROR

In this work, we consider a generalization of the expected calibration error to arbitrary $\ell_p$ norms.

**Definition 6.** Fix $p \geq 1$ and $k, \lambda \in \mathbb{N}^+$. Consider a $k$-class predictor $f : \mathcal{X} \to \Delta_k$ and a data distribution $D$ over features $\mathcal{X}$ and labels $[k]$. The $\ell_p$ calibration error of $f$ is defined as

$$\mathrm{Err}_p(f) := \left( \sum_{v \in V_\lambda^k} \sum_{j=1}^{k} (\mathrm{Err}(f, v, j))^p \right)^{1/p},$$

where $V_\lambda^k$ denotes the set of discretized bins,

$$\mathrm{Err}(f, v, j) := \left| \mathbb{E}_{(x,y) \sim D} \left[ (f(x)_j - y_j) \cdot \mathbb{I}\left[ R(f(x)) = v \right] \right] \right|$$

$$= \left| \mathbb{E}_{(x,y) \sim D} \left[ f(x)_j - y_j \mid R(f(x)) = v \right] \right| \mathbb{P}\left[ R(f(x)) = v \right]$$

measures the calibration error for bin $v$ and class $j$, and $y$ is the one-hot encoding of the label.

The special case when $p = 1$ corresponds to the expected calibration error (ECE), while the case when $p \to \infty$ corresponds to the calibration error considered by Haghtalab et al. (2023) and Dwork et al. (2023):

$$\max_{v \in V_\lambda^k, j \in [k]} \left| \mathbb{E}_{(x,y) \sim D} \left[ (f(x)_j - y_j) \cdot \mathbb{I}\left[ R(f(x)) = v \right] \right] \right|.$$

Our main result is a new algorithm that calibrates a given predictor $f$ to achieve $\ell_p$ calibration error of at most $\varepsilon$, using a polynomial number of samples for any $p > 1$. Furthermore, for $p = \infty$, the dependence of the algorithm's sample complexity on $\varepsilon$ is only $1/\varepsilon^2$ up to logarithmic factors, which is nearly optimal. The squared error of the calibrated predictor is lower than that of the original predictor, up to a small additive term introduced by discretization. Up to logarithmic factors, this additive term due to discretization is similar to the term in the previous work for binary predictors (Hébert-Johnson et al., 2018).

**Theorem 7.** *Fix* $p > 1$, $\varepsilon, \delta \in (0, 1)$ *and* $k \in \mathbb{N}^+$. *There exists an algorithm that takes as input a $k$-class predictor $f : \mathcal{X} \to \Delta_k$, and with probability at least $1 - \delta$ terminates after $O\left( \frac{2^{2/(p-1)}}{\varepsilon^{2p/(p-1)}} \right)$ time steps with total time polynomial in $k$ and $\frac{1}{\varepsilon}$. Using*

$$O\left( \left( \frac{2^{1/(p-1)}}{\varepsilon^{p/(p-1)}} \right)^2 \log^3 \left( \frac{2^{1/(p-1)}}{\varepsilon^{p/(p-1)}} \right) \log \left( \frac{2^{1/(p-1)} k}{\varepsilon^{p/(p-1)} \delta} \right) \right)$$

*samples from distribution $D$, it returns a $k$-class predictor $h : \mathcal{X} \to \Delta_k$ that has calibration error $\mathrm{Err}_p(h) \leq \varepsilon$ and squared error*

$$\mathbb{E}_D \left[ \| h(x) - y \|_2^2 \right] - \mathbb{E}_D \left[ \| f(x) - y \|_2^2 \right] \leq O\left( \frac{\varepsilon^{p/(p-1)}}{2^{1/(p-1)}} \log \left( \frac{2^{1/(p-1)}}{\varepsilon^{p/(p-1)}} \right) \right).$$

We present Algorithm 2 for calibrating a given $k$-class predictor $f$. The high-level structure of the algorithm, outlined in Algorithm 1, follows a standard approach in the literature. It first assigns datapoints to bins based on the level set of their rounded prediction $f(x)$, and then iteratively identifies groups of bins and classes with large calibration error, applying corrective updates as needed. At each time step $t$, to correct the prediction for a group of bins $S^{(t)}$ and class $j^{(t)}$ with large calibration error, the algorithm estimates the probability that datapoints in bins $S^{(t)}$ have label $j^{(t)}$. It then uses this estimate to correct the prediction vector for $S^{(t)}$ and projects the corrected vector onto the probability simplex $\Delta_k$ to ensure valid probability outputs, using this as the new prediction for datapoints assigned to $S^{(t)}$. If at time step $t$, there exists another group of bins $S'$ with prediction in the same level set as $S^{(t)}$, the algorithm merges these two groups. It assigns a single prediction vector to all the inputs in $S^{(t)} \cup S'$, selecting the prediction from whichever group has the largest estimated probability mass. However, merging bins may cause the estimation errors to accumulate, potentially leading to large calibration errors in the merged group. To mitigate this, the algorithm re-estimates the calibration error of each group after merging.

---

**Algorithm 1** Multiclass Calibration Outline

---

**Input:** predictor $f$

Discretize prediction space into bins and identify high-probability bins $B$

Create two parallel data structures:

    1. Estimation structure $M$ tracks statistics for groups of bins

    2. Prediction structure $G$ stores predictions and tracks calibration errors per group of bins

Initialize both structures, $M$ and $G$, to contain one group per high-probability bin in $B$

$t \leftarrow 0$

While there exists a group of bins in $G$ with large error for some class $j \in [k]$:

    Select group $S^{(t)} \in G$ and class $j^{(t)} \in [k]$ with large error

    Correct the prediction for $S^{(t)}$ and $j^{(t)}$

    Merge groups in $G$ with similar predictions to that of $S^{(t)}$

    Update structure $M$

    Estimate statistics and error for $S^{(t)}$

    $t \leftarrow t + 1$.

$h(x) = \begin{cases} \text{prediction for group } S \text{ in } G \text{ that contains } f(x) & \text{if } f(x) \text{ is in a high-probability bin} \\ \text{nearest valid probability vector to } f(x) & \text{o.w.} \end{cases}$

**Output:** calibrated predictor $h$

---

Our algorithm differs from existing binning-based calibration algorithms in two ways. First, it identifies a set of bins $B$ with large probability mass, because only such bins contribute significantly to the overall calibration error. The algorithm maintains a data structure $G$ containing disjoint groups of bins that may have large error and iteratively searches through them to identify groups requiring correction. Initially, $G$ contains a group for each high-probability bin. As the algorithm merges groups of bins, it updates $G$ accordingly. Second, the algorithm reduces the number of samples needed to estimate the calibration error by leveraging the fact that groups of bins are only merged over time and never split, and by applying Lemma 4 for adaptive data analysis. The groups of bins $S^{(t)}$ are selected adaptively, as their error depends on the current predictions. If we were to analyze the sample complexity using standard concentration inequalities, this adaptivity would require the use of fresh samples at every time step. To avoid this inefficiency, our algorithm maintains error estimates for $O(\log |B|)$ collections of evolving disjoint groups of bins, denoted collectively as $M$. Note that $M$ forms a partition of $B$. An interesting property of this structure is that any group of bins in $G$ for which we need to estimate the calibration error can be expressed as a disjoint union of groups in $M$. As a result, the calibration error estimate of $S^{(t)}$ can be computed efficiently by summing the estimates for groups in $M$ that are subsets of $S^{(t)}$. The sizes of the groups in $M$ are powers of 2 and all groups of the same size that arise during the execution of the algorithm remain disjoint. For each group size $2^i$ and each type of estimate, we maintain a separate pool of samples. Since a group in $M$ can contain at most $|B|$ distinct bins, we need $O(\log |B|)$ separate sample pools. We analyze the sample complexity after proving Lemma 9, which bounds the number of samples required to estimate a collection of disjoint, adaptively chosen queries.

We show that Algorithm 2 satisfies Theorem 7. The proof is presented step by step in the following three subsections, with key results organized into several lemmas. Lemmas 8 and 9 show that all estimated quantities are within small additive error of the true quantities. Lemmas 11, 12, and 13 provide a bound on the squared error of the modified predictor. Lemma 14 proves that the algorithm terminates after $O(2^{2/(p-1)}/\varepsilon^{2p/(p-1)})$ iterations, while Lemma 16 shows that the total runtime is polynomial in $1/\varepsilon$ and $k$. Finally, Lemma 15 establishes that the calibration error of the final predictor when the algorithm terminates is smaller than $\varepsilon$. All omitted proofs are provided in the Appendix.

### 3.1 CORRECTNESS OF ESTIMATES

In Algorithm 2 we use samples to compute three types of estimates. For the algorithm to function correctly, the estimates need to be sufficiently accurate. This requirement is captured by the following three events. Event $A_1$ ensures that $B$ contains bins with large probability masses. Events $A_2$ and $A_3$, together enable the algorithm to correctly adjust predictions and merge bins as needed.

---

**Algorithm 2** Multiclass Calibration

---

**Input:** predictor $f$, discretization function $R$, parameters $\varepsilon$ and $\delta$.

Set $\beta \leftarrow \varepsilon^{p/(p-1)} 2^{-1/(p-1)}$ and $\lambda \leftarrow \lceil 1/\beta \rceil$.
For all bins $v \in V_\lambda^k$:
    Estimate probability mass of bin $v$, $\hat{\mu}_v \approx \mathbb{P}[R(f(x)) = v]$
Select high-probability bins $B \leftarrow \{v : \hat{\mu}_v \geq \beta/6\}$

$M \leftarrow$ initialize with one group $\{v\}$ per high-probability bin $v$ in $B$
$G \leftarrow$ initialize with one group $\{v\}$ per high-probability bin $v$ in $B$
$t \leftarrow 0$
For each group $\{v\} \in M$:
    Estimate probability $\hat{P}_{\{v\}} \approx \mathbb{P}[R(f(x)) \in \{v\}]$
    Estimate mean label $\hat{E}_{\{v\},j} \approx \mathbb{E}_{(x,y)\sim D}[y_j \mathbb{I}[R(f(x)) \in \{v\}]]$ for all $j \in [k]$
For each group $\{v\} \in G$:
    $\text{pred}(\{v\}) \leftarrow \rho(v)$
    Compute $\hat{\text{Err}}(\{v\}, j) \leftarrow \left| \hat{P}_{\{v\}} \text{pred}(\{v\})_j - \hat{E}_{\{v\},j} \right|$ for each class $j \in [k]$

While $\exists$ group $S \in G$ with error $\hat{\text{Err}}(S, j) > \beta/2$ for some class $j \in [k]$:
    Select group $S^{(t)} \in G$ and class $j^{(t)} \in [k]$ with $\hat{\text{Err}}(S^{(t)}, j^{(t)}) > \beta/2$
    $z_{j^{(t)}}^{(t)} \leftarrow \min\left( \left(\sum_{S \in M: S \subseteq S^{(t)}} \hat{E}_{S,j^{(t)}}\right) / \left(\sum_{S \in M: S \subseteq S^{(t)}} \hat{P}_S\right), 1 \right)$
    For all other classes $j \neq j^{(t)}$: $z_j^{(t)} \leftarrow \text{pred}\left(S^{(t)}\right)_j$
    $\text{pred}\left(S^{(t)}\right) \leftarrow \pi\left(z^{(t)}\right)$
    If there exists group $S' \neq S^{(t)}$ in $G$ such that $R\left(\text{pred}\left(S'\right)\right) = R\left(\text{pred}\left(S^{(t)}\right)\right)$:
        Merge $S^{(t)}$ and $S'$ into a single group in $G$
        If $\sum_{S \in M: S \subseteq S^{(t)}} \hat{P}_S \leq \sum_{S \in M: S \subseteq S'} \hat{P}_S$:
            $\text{pred}\left(S^{(t)} \cup S'\right) \leftarrow \text{pred}\left(S'\right)$
        else:
            $\text{pred}\left(S^{(t)} \cup S'\right) \leftarrow \text{pred}\left(S^{(t)}\right)$
        $S^{(t)} \leftarrow S^{(t)} \cup S'$
    While there exist groups $S_1 \neq S_2$ in $M$ that are subsets of $S^{(t)}$ with the same cardinality:
        Merge $S_1$ and $S_2$ in $M$
        Estimate probability $\hat{P}_{S_1 \cup S_2} \approx \mathbb{P}[R(f(x)) \in S_1 \cup S_2]$
        Estimate mean label $\hat{E}_{S_1 \cup S_2, j} \approx \mathbb{E}_{(x,y)\sim D}[y_j \mathbb{I}[R(f(x)) \in S_1 \cup S_2]]$ for all $j \in [k]$
    Compute $\hat{\text{Err}}(S^{(t)}, j) \leftarrow \left| \left(\sum_{S \in M: S \subseteq S^{(t)}} \hat{P}_S\right) \text{pred}\left(S^{(t)}\right)_j - \sum_{S \in M: S \subseteq S^{(t)}} \hat{E}_{S,j} \right|, \forall j \in [k]$
    $t \leftarrow t + 1$.

$$h(x) = \begin{cases} \text{pred}(S), \text{ where } S \text{ is the group in } G \text{ that contains } R\left(f(x)\right) & \text{if } R\left(f(x)\right) \in B \\ \rho\left(R\left(f(x)\right)\right) & \text{o.w.} \end{cases}$$
**Output:** $h$

---

**Important Events:**

1. Event $A_1$: $|\hat{\mu}_v - \mathbb{P}[R(f(x)) = v]| \leq \frac{\beta}{12}, \forall v \in V_\lambda^k$.

2. Event $A_2$: $\left| \hat{P}_S - \mathbb{P}[R(f(x)) \in S] \right| \leq \frac{\beta}{36(\lfloor \log_2 |B| \rfloor + 1)}$, for all groups of bins $S$ in $M$ that ever occur during the execution of the algorithm.

3. Event $A_3$: $\left| \hat{E}_{S,j} - \mathbb{E}_{(x,y)\sim D}[y_j \mathbb{I}[R(f(x)) \in S]] \right| \leq \frac{\beta}{36(\lfloor \log_2 |B| \rfloor + 1)}$, for all groups of bins $S$ in $M$ that ever occur during the execution of the algorithm and all classes $j \in [k]$.

First, for every level set $v \in V_\lambda^k$ we estimate the probability that the rounded prediction of the given predictor $R(f(x))$ equals $v$. By Lemma 8, if we set $\alpha_1 = \beta/12$ and $\delta_1 = \delta/3$, we know that

using $O\left(\frac{1}{\beta}\log\left(\frac{|V_\lambda^k|}{\delta}\right) + \frac{1}{\beta^2}\log\left(\frac{1}{\beta\delta}\right)\right)$ samples we get estimates such that with probability at least $1 - \delta/3$

$$|\hat{\mu}_v - \mathbb{P}\left[R(f(x)) = v\right]| \le \frac{\beta}{12}, \ \forall v \in V_\lambda^k.$$

**Lemma 8.** *Fix $\delta_1, \alpha_1 \in (0,1)$. Using $O\left(\frac{1}{\alpha_1}\log\left(\frac{|V_\lambda^k|}{\delta_1}\right) + \frac{1}{\alpha_1^2}\log\left(\frac{1}{\alpha_1\delta_1}\right)\right)$ samples, we can estimate $\hat{\mu}_v$, for all $v \in V_\lambda^k$, s.t. with probability at least $1 - \delta_1$*

$$|\hat{\mu}_v - \mathbb{P}\left[R(f(x)) = v\right]| \le \alpha_1, \ \forall v \in V_\lambda^k.$$

For every group of bins $S$ that appears in $M$ during the execution of the algorithm, we estimate two types of quantities: the probability that the prediction $R(f(x))$ is in one of the bins in $S$ and the expected label $y_j$ of points $(x,y)$ whose prediction $R(f(x))$ is in one of the bins in $S$, for all $j \in [k]$. The sizes of groups in $M$ are all powers of 2 and all groups of the same size that occur during the execution of the algorithm are disjoint. For each group size $2^i$ and for each type of estimate, probability or expected label, we maintain a separate pool of samples. Since there can be at most $|B|$ distinct bins in a group in $M$, we need $O(\log|B|)$ separate sample pools. To analyze the sample complexity, we apply the adaptive data analysis result of Lemma 9 because the algorithm picks the set that needs adjustment adaptively at each time step.

**Lemma 9.** *Fix $n, k \in \mathbb{N}^+$ and $\alpha, \delta \in (0,1)$. Consider an adaptive algorithm $A$, a distribution $D$ over the domain $\mathcal{X} \times \mathcal{Y}$, and a function $\phi : \mathcal{X} \times \mathcal{Y} \to \Delta_k$. The algorithm adaptively selects a sequence of $n$ disjoint events for $D$ as follows. First, it selects $E_1$ and estimates $\mathbb{E}_{(x,y)\sim D}\left[\phi(x,y)_j \cdot \mathbb{I}\left[(x,y) \in E_1\right]\right]$, for all $j \in [k]$. Then, it selects event $E_2$, disjoint from $E_1$, and estimates $\mathbb{E}_{(x,y)\sim D}\left[\phi(x,y)_j \cdot \mathbb{I}\left[(x,y) \in E_2\right]\right]$, for all $j \in [k]$, and so on. With $O\left(\frac{\log(nk/\delta)}{\alpha^2}\right)$ shared samples, we can estimate all expectations up to additive error $\alpha$ and failure probability $\delta$.*

By Lemma 9, we get that for a fixed group size $2^i \le |B|$, using $O\left(\frac{\log^2(|B|)\log(|B|\log|B|/\delta)}{\beta^2}\right)$ samples we get probability estimates such that with probability at least $1 - \frac{\delta}{3(\lfloor\log_2|B|\rfloor+1)}$

$$\left|\hat{P}_S - \mathbb{P}\left[R(f(x)) \in S\right]\right| \le \frac{\beta}{36(\lfloor\log_2|B|\rfloor + 1)},$$

for all groups of bins $S$ in $M$ of size $2^i$ that ever occur during the execution of the algorithm. Similarly, by Lemma 9 we get that for a fixed group size $2^i \le |B|$, using $O\left(\frac{\log^2(|B|)\log(|B|k\log|B|/\delta)}{\beta^2}\right)$, samples we get expected label estimates such that with probability at least $1 - \frac{\delta}{3(\lfloor\log_2|B|\rfloor+1)}$

$$\left|\hat{E}_{S,j} - \mathbb{E}_{(x,y)\sim D}\left[y_j\mathbb{I}\left[R(f(x)) \in S\right]\right]\right| \le \frac{\beta}{36(\lfloor\log_2|B|\rfloor + 1)},$$

for all groups of bins $S$ in $M$ of size $2^i$ that ever occur during the execution of the algorithm and all classes $j \in [k]$.

The number of groups with different sizes up to $|B|$ that are powers of 2 is at most $\lfloor\log_2|B|\rfloor + 1$. Thus, we have that

$$\mathbb{P}\left[\neg A_1 \text{ or } \neg A_2 \text{ or } \neg A_3\right] \le \mathbb{P}[\neg A_1] + \mathbb{P}[\neg A_2] + \mathbb{P}[\neg A_3]$$

$$\le \frac{\delta}{3} + (\lfloor\log_2|B|\rfloor + 1)\frac{\delta}{3(\lfloor\log_2|B|\rfloor + 1)} + (\lfloor\log_2|B|\rfloor + 1)\frac{\delta}{3(\lfloor\log_2|B|\rfloor + 1)} \le \delta$$

If event $A_1$ is true, then the size of $|B|$ is at most $O\left(\frac{1}{\beta}\right)$ because $B = \left\{v : v \in V_\lambda^k, \hat{\mu}_v \ge \beta/6\right\}$ and $\sum_{v \in V_\lambda^k}\mathbb{P}\left[R(f(x)) = v\right] = 1$. Thus, the algorithm can use

$O\left(\frac{1}{\beta}\log\left(\frac{|V_\lambda^k|}{\delta}\right)+\frac{1}{\beta^2}\log^3\left(\frac{1}{\beta}\right)\log\left(\frac{k\log(1/\beta)}{\beta\delta}\right)\right)$ samples in total. Lemma 3 provides a bound of the size of $V_\lambda^k$.

To estimate the probability of a group of bins $S \in G$, we compute the sum of probability estimates for all subsets $S' \subseteq S$ that are in $M$ and use the following Lemma to bound the overall error. We estimate the expected label in a similar way.

**Lemma 10.** *For each $S \in G$, the number of subsets $S' \in M$ such that $S' \subseteq S$ is at most $O\left(\log|B|\right)$.*

### 3.2 ACCURACY OF THE CALIBRATED PREDICTOR

In this subsection, we show that if the estimates are accurate, then Algorithm 2 constructs a multiclass predictor whose squared error is lower than that of the given predictor, up to a small additive term introduced by discretization. At each round $t$ before the algorithm terminates, it selects a bin $S^{(t)}$ and a coordinate $j^{(t)}$ with high calibration error. The algorithm then updates the predictor in two stages. In Stage 1, it computes an improved prediction vector $z^{(t)}$ for the selected bin and projects it to the simplex to obtain pred $\left(S^{(t)}\right)$. In Stage 2, it checks if there is another group $S'$ that gets mapped to the same level set as $S^{(t)}$ and if so it merges $S'$ and $S^{(t)}$. We analyze the change in the squared error at each time step by examining separately the change due to Stage 1 and Stage 2. Notably, in Lemma 12 we show that the squared error always decreases in Stage 1, whereas in Lemma 11 we demonstrate that Stage 2 might lead to a small increase. In both lemmas, we assume that the all the estimated quantities are accurate, meaning that events $A_1$, $A_2$ and $A_3$ as defined in the previous subsection hold. Lemma 13 provides an upper on the squared error due to the discretization of $f$.

For the purposes of this proof we define

$$h_t(x) = \begin{cases} \text{pred}(S), \text{where } S \text{ in } G \text{ contains } R\left(f(x)\right) \text{ at time step } t & \text{if } R\left(f(x)\right) \in B \\ \rho\left(R\left(f(x)\right)\right) & \text{o.w.} \end{cases}$$

**Lemma 11.** *Assuming that $A_1$, $A_2$ and $A_3$ hold, after $T$ time steps of the algorithm, the squared error of the predictor $h$ is*

$$\mathbb{E}\left[\|h(x) - y\|_2^2\right]$$

$$\leq \mathbb{E}\left[\|h_0(x) - y\|_2^2\right] + O\left(\beta\log\left(\frac{1}{\beta}\right)\right)$$

$$+ \sum_{t=0}^{T-1} \mathbb{E}\left[\|\pi(z^{(t)}) - y\|_2^2 - \|h_t(x) - y\|_2^2 \, \Big| \, R\left(f(x)\right) \in S^{(t)}\right] \mathbb{P}\left[R\left(f(x)\right) \in S^{(t)}\right].$$

**Lemma 12.** *Assuming that $A_1$, $A_2$ and $A_3$ hold, at time step $t$ of the algorithm*

$$\mathbb{E}\left[\|\pi(z^{(t)}) - y\|_2^2 - \|h_t(x) - y\|_2^2 \, \Big| \, R\left(f(x)\right) \in S^{(t)}\right] \mathbb{P}\left[R\left(f(x)\right) \in S^{(t)}\right] \leq -\beta^2/9.$$

**Lemma 13.** *The squared error at time step $0$ is $\mathbb{E}\left[\|h_0(x) - y\|_2^2\right] \leq \mathbb{E}\left[\|f(x) - y\|_2^2\right] + O(\beta)$.*

### 3.3 TERMINATION OF THE ALGORITHM WITH SMALL CALIBRATION ERROR

In this subsection, we show that, assuming that the estimates are accurate, the algorithm terminates after $O\left(1/\beta^2\right)$ steps with $\ell_p$ calibration error at most $O\left(\beta^{(p-1)/p}\right)$. Moreover, its total runtime is polynomial in $1/\beta$ and $k$.

**Lemma 14.** *Assuming that $A_1$, $A_2$ and $A_3$ hold, the algorithm terminates after at most $O\left(1/\beta^2\right)$ time steps.*

**Lemma 15.** *Assuming that $A_1$, $A_2$ and $A_3$ hold, the $\ell_p$ calibration error $(Err_p(h))^p$ is bounded by $O(\beta^{p-1})$.*

**Lemma 16.** *Assuming that $A_1$, $A_2$ and $A_3$ hold, the algorithm terminates in time $O\left(\frac{k}{\beta^2}\log^3\left(\frac{1}{\beta}\right)\log\left(\frac{k}{\beta\delta}\right)\right)$.*

Combining the results of Subsections 3.1, 3.2, and 3.3, we obtain the proof of Theorem 7.

## 4 CONCLUSION

In this work, we introduced the $\ell_p$ calibration error for multiclass predictors and presented an algorithm that modifies a given predictor to achieve low calibration error while preserving its accuracy using only a polynomial number of samples in the number of classes. The algorithm can be applied to any value of $p > 1$ and improves the known sample complexity in the case of $p = \infty$.

Related work in this area has explored multicalibration, where the calibration guarantees hold for many, possibly overlapping, populations. While our work focuses on calibration, an interesting direction for future research is to generalize our results to obtain stronger sample complexity in that setting as well.

## ETHICS STATEMENT

Our work advances the theoretical understanding of calibration for multiclass predictors. A practical implementation of the algorithm could be applied to real-world models to improve their reliability and interpretability. This could support efforts to responsibly deploy machine learning model systems in societal applications

## REPRODUCIBILITY STATEMENT

Our work is theoretical. The complete proofs of the lemmas and the main theorem can be found in Section 3 and the Appendix.

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

# A    APPENDIX

## A.1    PROOFS FROM SECTION 2

**Lemma 17** (Lemma 3 restated). *For any $\lambda, k \in \mathbb{N}^+$, the number of level sets in $V_\lambda^k$ is at most $\binom{\lambda+k}{k}$. Note that $\log\left(\left|V_\lambda^k\right|\right) = O\left(\min\left(k, \lambda\right) \ln\left(k + \lambda\right)\right)$ whereas $\log\left(\left|L^k\right|\right) = O\left(k \ln\left(\lambda\right)\right)$.*

*Proof.* Every $v \in V_\lambda^k$ corresponds to a $u \in \Delta_k$. Therefore, we have that

$$\sum_{i\in[k]} v_i = \sum_{i\in[k]} \frac{\lfloor u_i \lambda \rfloor}{\lambda} = 1 - \left(1 - \sum_{i\in[k]} \frac{\lfloor u_i \lambda \rfloor}{\lambda}\right).$$

Let $v_{k+1} = 1 - \sum_{i\in[k]} \frac{\lfloor u_i \lambda \rfloor}{\lambda}$, which is a non-negative integer multiple of $1/\lambda$. By rearranging the terms, we have that $\sum_{i\in[k+1]} v_i = 1$. The number of $k + 1$ tuples of non-negative integer multiples of $1/\lambda$ that sum up to $1$ is $\binom{\lambda+k}{k}$. Therefore, $\left|V_\lambda^k\right| = \binom{\lambda+k}{k}$. □

## A.2    PROOFS FROM SUBSECTION 3.1

**Lemma 18** (Lemma 8 restated). *Fix $\delta_1, \alpha_1 \in (0,1)$. Using $O\left(\frac{1}{\alpha_1} \log\left(\frac{|V_\lambda^k|}{\delta_1}\right) + \frac{1}{\alpha_1^2} \log\left(\frac{1}{\alpha_1 \delta_1}\right)\right)$ samples, we can estimate $\hat{\mu}_v$, for all $v \in V_\lambda^k$, s.t. with probability at least $1 - \delta_1$*

$$|\hat{\mu}_v - \mathbb{P}\left[R(f(x)) = v\right]| \le \alpha_1, \ \forall v \in V_\lambda^k.$$

*Proof.* There are at most $\frac{1}{\alpha_1}$ bins such that $\mathbb{P}[R(f(x)) = v] \geq \alpha_1$. We show that using $m_1 = \frac{1}{2\alpha_1^2} \ln\left(\frac{4}{\alpha_1 \delta_1}\right)$ samples, we can estimate all of them up to additive error $\alpha_1$. By applying the Hoeffding inequality and a union bound we obtain that

$$\mathbb{P}\left[\exists v \text{ s.t. } \mathbb{P}[R(f(x)) = v] \geq \alpha_1 : |\hat{\mu}_v - \mathbb{P}[R(f(x)) = v]| \geq \alpha_1\right]$$
$$\leq \frac{2|\{v : \mathbb{P}[R(f(x)) = v] \geq \alpha_1\}|}{e^{2\alpha_1^2 m_1}}$$
$$\leq \frac{2}{\alpha_1 e^{2\alpha_1^2 m_1}} \leq \frac{\delta_1}{2}.$$

For the rest of the bins whose probabilities are less than $\alpha_1$, we show that using $m_2 = \frac{4}{3\alpha_1} \ln\left(2|V_\lambda^k|/\delta_1\right)$ samples is enough to estimate all of them up to additive error $\alpha_1$. In this case, we have that for all $v$ such that $\mathbb{P}[R(f(x)) = v] < \alpha_1$, $\mathbb{P}[R(f(x)) = v] - \hat{\mu}_v < \alpha_1$. By applying Lemma 5 we also get that

$$\mathbb{P}\left[\exists v \text{ s.t. } \mathbb{P}[R(f(x)) = v] < \alpha_1 : \hat{\mu}_v - \mathbb{P}[R(f(x)) = v] \geq \alpha_1\right]$$
$$\leq |V_\lambda^k| \cdot \exp\left(-\frac{m_2 \alpha_1^2}{2(\alpha_1 + \alpha_1/3)}\right) \leq \frac{\delta_1}{2}.$$

By union bound we obtain that if we use $O\left(\frac{1}{\alpha_1} \log\left(\frac{|V_\lambda^k|}{\delta_1}\right) + \frac{1}{\alpha_1^2} \log\left(\frac{1}{\alpha_1 \delta_1}\right)\right)$ samples, then

$$\mathbb{P}\left[\exists v \in V_\lambda^k : |\hat{\mu}_v - \mathbb{P}[R(f(x)) = v]| \geq \alpha_1\right] \leq \delta_1. \qquad \square$$

**Lemma 19** (Lemma 9 restated). *Fix $n, k \in \mathbb{N}^+$ and $\alpha, \delta \in (0, 1)$. Consider an adaptive algorithm $A$, a distribution $D$ over the domain $\mathcal{X} \times \mathcal{Y}$, and a function $\phi : \mathcal{X} \times \mathcal{Y} \to \Delta_k$. The algorithm adaptively selects a sequence of $n$ disjoint events for $D$ as follows. First, it selects $E_1$ and estimates $\mathbb{E}_{(x,y) \sim D}[\phi(x, y)_j \cdot \mathbb{I}[(x, y) \in E_1]]$, for all $j \in [k]$. Then, it selects event $E_2$, disjoint from $E_1$, and estimates $\mathbb{E}_{(x,y) \sim D}[\phi(x, y)_j \cdot \mathbb{I}[(x, y) \in E_2]]$, for all $j \in [k]$, and so on. With $O\left(\frac{\log(nk/\delta)}{\alpha^2}\right)$ shared samples, we can estimate all expectations up to additive error $\alpha$ and failure probability $\delta$.*

*Proof.* There are many ways to achieve this. Here, we describe one approach using differential privacy and a transfer theorem to adaptive analysis. The algorithm uses a set $S$ of $m = \frac{32 \ln(4nk/\delta)}{\alpha^2}$ samples and for each event $E_i$ and coordinate $j \in [k]$, it reports $\hat{e}_{i,j} = \frac{1}{m} \sum_{u \in S} \phi(u)_j \cdot \mathbb{I}[u \in E_i] + \varepsilon_{i,j}$, where $\varepsilon_{i,j} \sim \text{Lap}(8/(m\alpha))$. Because the events are disjoint and each sample contributes to at most one event, the $\ell_1$ global sensitivity of the $k \times n$-dimensional vector $(e_{1,1}, \ldots, e_{1,k}, \ldots, e_{n,1}, \ldots, e_{n,k})$, where $e_{i,j} = \frac{1}{m} \sum_{u \in S} \phi(u)_j \cdot \mathbb{I}[u \in E_i]$, is at most $2/m$. Hence, algorithm $A$ is $(\alpha/4, 0)$-differentially private. Since $\varepsilon_{1,1}, \ldots, \varepsilon_{n,k}$ are i.i.d. Laplace random variables with $\lambda = \frac{8}{m\alpha}$, we know that for any $t > 0$, $\mathbb{P}\left[\max_{i \in [n], j \in [k]} |\varepsilon_{i,j}| > t\lambda\right] \leq nde^{-t}$. For $t = \ln(2nk/\delta)$, we get that with probability at least $1 - \frac{\delta}{2}$, the maximum additive error $|\varepsilon_{i,j}|$ is at most $\frac{8 \ln(2nk/\delta)}{m\alpha}$. By Lemma 4, with probability at least $1 - \delta$, we have that

$$\max_{i \in [n], j \in [d]} \left|\mathbb{E}_{(x,y) \sim D}[\phi(x, y)_j \cdot \mathbb{I}[(x, y) \in E_i]] - \hat{e}_{i,j}\right| \leq \frac{8 \ln\left(\frac{2nk}{\delta}\right)}{m\alpha} + e^{\alpha/4} - 1 + \sqrt{\frac{2 \ln\left(\frac{4}{\delta}\right)}{m}}$$
$$\leq \frac{\alpha}{4} + \frac{\alpha}{2} + \frac{\alpha}{4} = \alpha.$$

$$\square$$

**Lemma 20** (Lemma 10 restated). *For each $S \in G$, the number of subsets $S' \in M$ such that $S' \subseteq S$ is at most $O(\log|B|)$.*

*Proof.* For a fixed $S \in G$, all $S' \in M$ such that $S' \subseteq S$ are of different sizes. This holds because if there were two subsets $S_1, S_2 \in M$ such that $S_1, S_2 \subseteq S$ and $|S_1| = |S_2|$, we would have already merged them. Additionally, the sizes of all $S' \in M$ are powers of 2. The number of sets with different sizes up to $|B|$ that are powers of 2 is at most $\lfloor \log_2 |B| \rfloor + 1$. □

## A.3 PROOFS FROM SUBSECTION 3.2

**Lemma 21** (Lemma 11 restated). *Assuming that $A_1$, $A_2$ and $A_3$ hold, after $T$ time steps of the algorithm, the squared error of the predictor $h$ is*

$$
\mathbb{E}\left[\|h(x) - y\|_2^2\right]
$$

$$
\leq \mathbb{E}\left[\|h_0(x) - y\|_2^2\right] + O\left(\beta \log\left(\frac{1}{\beta}\right)\right)
$$

$$
+ \sum_{t=0}^{T-1} \mathbb{E}\left[\|\pi(z^{(t)}) - y\|_2^2 - \|h_t(x) - y\|_2^2 \,\Big|\, R(f(x)) \in S^{(t)}\right] \mathbb{P}\left[R(f(x)) \in S^{(t)}\right].
$$

*Proof.* At each time step $t \leq T - 1$ there are three possible cases depending on whether and how the algorithm merges bins after updating the prediction for $S^{(t)}$.

Case 1: there is no $S'$ such that $R\left(\pi\left(z^{(t)}\right)\right) = R(\text{pred}(S'))$. Then,

$$
\mathbb{E}\left[\|h_{t+1}(x) - y\|_2^2\right] - \mathbb{E}\left[\|h_t(x) - y\|^2\right]
$$

$$
= \mathbb{E}\left[\|h_{t+1}(x) - y\|_2^2 - \|h_t(x) - y\|_2^2 \,\Big|\, R(f(x)) \in S^{(t)}\right] \mathbb{P}\left[R(f(x)) \in S^{(t)}\right]
$$

$$
= \mathbb{E}\left[\left\|\pi\left(z^{(t)}\right) - y\right\|_2^2 - \|h_t(x) - y\|_2^2 \,\Big|\, R(f(x)) \in S^{(t)}\right] \mathbb{P}\left[R(f(x)) \in S^{(t)}\right].
$$

Case 2: there is a $S'$ such that $R\left(\pi\left(z^{(t)}\right)\right) = R(\text{pred}(S'))$ and $\sum_{S \in M : S \subseteq S^{(t)}} \hat{P}_S > \sum_{S \in M : S \subseteq S'} \hat{P}_S$. Then,

$$
\mathbb{E}\left[\|h_{t+1}(x) - y\|_2^2\right] - \mathbb{E}\left[\|h_t(x) - y\|_2^2\right]
$$

$$
= \mathbb{E}\left[\left\|\pi\left(z^{(t)}\right) - y\right\|_2^2 - \|h_t(x) - y\|_2^2 \,\Big|\, R(f(x)) \in S^{(t)}\right] \mathbb{P}\left[R(f(x)) \in S^{(t)}\right]
$$

$$
+ \mathbb{E}\left[\left\|\pi\left(z^{(t)}\right) - y\right\|_2^2 - \|h_t(x) - y\|_2^2 \,\Big|\, R(f(x)) \in S'\right] \mathbb{P}[R(f(x)) \in S']
$$

$$
\leq \mathbb{E}\left[\left\|\pi\left(z^{(t)}\right) - y\right\|_2^2 - \|h_t(x) - y\|_2^2 \,\Big|\, R(f(x)) \in S^{(t)}\right] \mathbb{P}\left[R(f(x)) \in S^{(t)}\right]
$$

$$
+ \frac{4}{\lambda} \mathbb{P}[R(f(x)) \in S'].
$$

The last inequality holds because if $R\left(f\left(x\right)\right) \in S'$, we have that

$$\mathbb{E}\left[\left\|\pi\left(z^{(t)}\right) - y\right\|_2^2 - \|h_t\left(x\right) - y\|_2^2 \,\middle|\, R\left(f\left(x\right)\right) \in S'\right]$$

$$= \mathbb{E}\left[\left\|\pi\left(z^{(t)}\right) - y\right\|_2^2 - \|\mathrm{pred}\left(S'\right) - y\|_2^2 \,\middle|\, R\left(f\left(x\right)\right) \in S'\right]$$

$$\leq \left\|\pi\left(z^{(t)}\right)\right\|_2^2 - \|\mathrm{pred}\left(S'\right)\|_2^2 + 2\max_{j \in [k]}\left|\pi\left(z^{(t)}\right)_j - \mathrm{pred}\left(S'\right)_j\right|$$

$$\leq \left(\max_{j \in [k]}\left|\pi\left(z^{(t)}\right)_j - \mathrm{pred}\left(S'\right)_j\right|\right)\sum_{j \in [k]}\left(\left|\pi\left(z^{(t)}\right)_j\right| + \left|\mathrm{pred}\left(S'\right)_j\right|\right)$$

$$+ 2\max_{j \in [k]}\left|\pi\left(z^{(t)}\right)_j - \mathrm{pred}\left(S'\right)_j\right|.$$

Since both $\pi\left(z^{(t)}\right)$ and $\mathrm{pred}\left(S'\right)$ are in the same level set when rounded by $R$, for each coordinate $j \in [k]$, $\left|\pi\left(z^{(t)}\right)_j - \mathrm{pred}\left(S'\right)_j\right| \leq 1/\lambda$. Furthermore, both $\pi\left(z^{(t)}\right)$ and $\mathrm{pred}\left(S'\right)$ are probability distributions and, hence, their coordinates sum to $1$. Therefore,

$$\left(\max_{j \in [k]}\left|\pi\left(z^{(t)}\right)_j - \mathrm{pred}\left(S'\right)_j\right|\right)\sum_{j \in [k]}\left(\left|\pi\left(z^{(t)}\right)_j\right| + \left|\mathrm{pred}\left(S'\right)_j\right|\right) \leq \frac{2}{\lambda}.$$

Case 3: there is a $S'$ such that $R\left(\pi\left(z^{(t)}\right)\right) = R\left(\mathrm{pred}\left(S'\right)\right)$ and $\sum_{S \in M : S \subseteq S^{(t)}}\hat{P}_S \leq \sum_{S \in M : S \subseteq S'}\hat{P}_S$. Then,

$$\mathbb{E}\left[\|h_{t+1}\left(x\right) - y\|_2^2\right] - \mathbb{E}\left[\|h_t\left(x\right) - y\|_2^2\right]$$

$$= \mathbb{E}\left[\|\mathrm{pred}\left(S'\right) - y\|_2^2 - \|h_t\left(x\right) - y\|_2^2 \,\middle|\, R\left(f\left(x\right)\right) \in S^{(t)}\right]\mathbb{P}\left[R\left(f\left(x\right)\right) \in S^{(t)}\right]$$

$$= \mathbb{E}\left[\left\|\pi\left(z^{(t)}\right) - y\right\|_2^2 - \|h_t\left(x\right) - y\|_2^2 \,\middle|\, R\left(f\left(x\right)\right) \in S^{(t)}\right]\mathbb{P}\left[R\left(f\left(x\right)\right) \in S^{(t)}\right]$$

$$+ \mathbb{E}\left[\|\mathrm{pred}\left(S'\right) - y\|_2^2 - \left\|\pi\left(z^{(t)}\right) - y\right\|_2^2 \,\middle|\, R\left(f\left(x\right)\right) \in S^{(t)}\right]\mathbb{P}\left[R\left(f\left(x\right)\right) \in S^{(t)}\right]$$

$$\leq \mathbb{E}\left[\left\|\pi\left(z^{(t)}\right) - y\right\|_2^2 - \|h_t\left(x\right) - y\|_2^2 \,\middle|\, R\left(f\left(x\right)\right) \in S^{(t)}\right]\mathbb{P}\left[R\left(f\left(x\right)\right) \in S^{(t)}\right]$$

$$+ \frac{4}{\lambda}\mathbb{P}\left[R\left(f\left(x\right)\right) \in S^{(t)}\right].$$

Similary to the previous case, the last inequality holds because we have that

$$\mathbb{E}\left[\|\mathrm{pred}\left(S'\right) - y\|_2^2 - \left\|\pi\left(z^{(t)}\right) - y\right\|_2^2 \,\middle|\, R\left(f\left(x\right)\right) \in S^{(t)}\right]$$

$$\leq \|\mathrm{pred}\left(S'\right)\|_2^2 - \left\|\pi\left(z^{(t)}\right)\right\|_2^2 + 2\max_{j \in [k]}\left|\pi\left(z^{(t)}\right)_j - \mathrm{pred}\left(S'\right)_j\right|$$

$$\leq \left(\max_{j \in [k]}\left|\mathrm{pred}\left(S'\right)_j - \pi\left(z^{(t)}\right)_j\right|\right)\sum_{j \in [k]}\left(\left|\mathrm{pred}\left(S'\right)_j\right| + \left|\pi\left(z^{(t)}\right)_j\right|\right)$$

$$+ 2\max_{j \in [k]}\left|\pi\left(z^{(t)}\right)_j - \mathrm{pred}\left(S'\right)_j\right|$$

$$\leq \frac{4}{\lambda}.$$

In all three cases discussed above, the upper bound includes the term

$$\mathbb{E}\left[\left\|\pi\left(z^{(t)}\right) - y\right\|_2^2 - \|h_t\left(x\right) - y\|_2^2 \,\middle|\, R\left(f\left(x\right)\right) \in S^{(t)}\right]\mathbb{P}\left[R\left(f\left(x\right)\right) \in S^{(t)}\right].$$

We can interpret the merge in Stage 2 in two ways depending on the case. In Case 2, the algorithm moves the prediction of $S'$ from $\text{pred}(S')$ to $\pi\left(z^{(t)}\right)$. In Case 3, it moves the prediction of $S^{(t)}$ from $\pi\left(z^{(t)}\right)$ to $\text{pred}(S')$. By summing the squared error differences over all time steps $t = 0$ to $T$, we get that

$$\mathbb{E}\left[\|h_T(x) - y\|_2^2\right] - \mathbb{E}\left[\|h_0(x) - y\|_2^2\right]$$

$$\leq \sum_{t=0}^{T-1} \mathbb{E}\left[\left\|\pi\left(z^{(t)}\right) - y\right\|_2^2 - \|h_t(x) - y\|_2^2 \,\middle|\, R(f(x)) \in S^{(t)}\right] \mathbb{P}\left[R(f(x)) \in S^{(t)}\right]$$

$$+ \frac{4}{\lambda} \sum_{t=0}^{T-1} \mathbb{P}\left[R(f(x)) \text{ is in the bin moved in Stage 2 of round } t\right].$$

Let $\tau(v)$ denote the number of times the level set $v$ is in the bin whose prediction gets moved in Stage 2. Then, $\sum_{t=0}^{T-1} \mathbb{P}[R(f(x)) \text{ is in the bin moved in Stage 2 of round } t] = \sum_{v \in B} \mathbb{P}[R(f(x)) = v] \cdot \tau(v)$.

We now establish an upper bound on $\tau(v)$ for $v \in B$. Suppose that $v$ is in the bin that gets moved in Stage 2 of some time step $t$, during the merge bins $S_a$ and $S_b$. Without loss of generality, assume that $S_a$ is the bin being moved. This implies that $v \in S_a$ and $\sum_{S \in M : S \subseteq S_a} \hat{P}_S \leq \sum_{S \in M : S \subseteq S_b} \hat{P}_S$. By the accuracy of the probability estimates, we have that $\mathbb{P}[R(f(x)) \in S_a] \leq \mathbb{P}[R(f(x)) \in S_b] + \beta/18$. Since $S_a$ and $S_b$ are disjoint, $\mathbb{P}[R(f(x)) \in S_a \cup S_b] \geq \mathbb{P}[R(f(x)) \in S_a] - \beta/18$. Since each merge involving moving the bin with $v$ (almost) doubles the size of the bin containing it, we have that

$$2^{\tau(v)} \mathbb{P}[R(f(x)) = v] - \frac{\beta}{36} \sum_{i=1}^{\tau(v)} 2^i \leq 1.$$

Hence,

$$\tau(v) \leq \log_2\left(\frac{1 - \beta/18}{\mathbb{P}[R(f(x)) = v] - \beta/18}\right).$$

Since $\varepsilon < 1$, we have $\beta = \varepsilon^{p/(p-1)} \cdot 2^{-1/(p-1)} < 1$. Additionally, $\mathbb{P}[R(f(x)) = v] \geq \beta/6 - \beta/12 = \beta/12$ because $v \in B$. Therefore, $\tau(v) \leq \log_2(36/\beta)$. Since $\lambda = \lceil 1/\beta \rceil$, we conclude that

$$\mathbb{E}\left[\|h_T(x) - y\|_2^2\right] - \mathbb{E}\left[\|h_0(x) - y\|_2^2\right]$$

$$\leq \sum_{t=0}^{T-1} \mathbb{E}\left[\left\|\pi\left(z^{(t)}\right) - y\right\|_2^2 - \|h_t(x) - y\|_2^2 \,\middle|\, R(f(x)) \in S^{(t)}\right] \mathbb{P}\left[R(f(x)) \in S^{(t)}\right]$$

$$+ \frac{4}{\lceil 1/\beta \rceil} \log_2\left(\frac{36}{\beta}\right).$$

$$\square$$

**Lemma 22** (Lemma 12 restated). *Assuming that $A_1$, $A_2$ and $A_3$ hold, at time step $t$ of the algorithm*

$$\mathbb{E}\left[\|\pi(z^{(t)}) - y\|_2^2 - \|h_t(x) - y\|_2^2 \,\middle|\, R(f(x)) \in S^{(t)}\right] \mathbb{P}\left[R(f(x)) \in S^{(t)}\right] \leq -\beta^2/9.$$

*Proof.* At each time step $t \leq T-1$, before the algorithm terminates we observe the following. Since $\pi\left(z^{(t)}\right) = \arg\min_{v \in \Delta_k}\left\|v - z^{(t)}\right\|_2$ and $y \in \Delta_k$, we have that $\left\|\pi\left(z^{(t)}\right) - y\right\|_2 \leq \left\|z^{(t)} - y\right\|_2$. Therefore, it suffices to find an upper bound for the following quantity:

$$\mathbb{E}\left[\left\|z^{(t)} - y\right\|_2^2 - \|h_t(x) - y\|_2^2 \,\middle|\, R(f(x)) \in S^{(t)}\right] \mathbb{P}\left[R(f(x)) \in S^{(t)}\right].$$

For simplicity, let $u^{(t)} = \mathrm{pred}\left(S^{(t)}\right)$ denote the previous prediction for group $S^{(t)}$. Then we have that

$$
\mathbb{E}\left[\left\|z^{(t)} - y\right\|_2^2 - \left\|u^{(t)} - y\right\|_2^2 \middle| R\left(f\left(x\right)\right) \in S^{(t)}\right] \mathbb{P}\left[R\left(f\left(x\right)\right) \in S^{(t)}\right]
$$

$$
= \mathbb{E}\left[\left(z_{j^{(t)}}^{(t)} - y_{j^{(t)}}\right)^2 - \left(u_{j^{(t)}}^{(t)} - y_{j^{(t)}}\right)^2 \middle| R\left(f\left(x\right)\right) \in S^{(t)}\right] \mathbb{P}\left[R\left(f\left(x\right)\right) \in S^{(t)}\right]
$$

$$
= \left(\left(z_{j^{(t)}}^{(t)}\right)^2 - \left(u_{j^{(t)}}^{(t)}\right)^2\right) \mathbb{P}\left[R\left(f\left(x\right)\right) \in S^{(t)}\right] + \left(2u_{j^{(t)}}^{(t)} - 2z_{j^{(t)}}^{(t)}\right) \mathbb{E}\left[y_{j^{(t)}} \mathbb{I}\left[R\left(f\left(x\right)\right) \in S^{(t)}\right]\right]
$$

$$
= \left(z_{j^{(t)}}^{(t)} - u_{j^{(t)}}^{(t)}\right) \left(\left(z_{j^{(t)}}^{(t)} + u_{j^{(t)}}^{(t)}\right) \mathbb{P}\left[R\left(f\left(x\right)\right) \in S^{(t)}\right] - 2\mathbb{E}\left[y_{j^{(t)}} \mathbb{I}\left[R\left(f\left(x\right)\right) \in S^{(t)}\right]\right]\right).
$$

The value of $z_{j^{(t)}}^{(t)}$, as assigned by the algorithm, falls into one of two cases. Simultaneously, we have bounds on the value of $u_{j^{(t)}}^{(t)}$, since the algorithm has selected a bin $S^{(t)}$ with large error. These bounds play a crucial role in analyzing

$$
\left(z_{j^{(t)}}^{(t)} - u_{j^{(t)}}^{(t)}\right)
$$

and

$$
\left(\left(z_{j^{(t)}}^{(t)} + u_{j^{(t)}}^{(t)}\right) \mathbb{P}\left[R\left(f\left(x\right)\right) \in S^{(t)}\right] - 2\mathbb{E}\left[y_{j^{(t)}} \mathbb{I}\left[R\left(f\left(x\right)\right) \in S^{(t)}\right]\right]\right).
$$

Case 1: $z_{j^{(t)}}^{(t)} = 1$. Then, $\sum_{S \in M : S \subseteq S^{(t)}} \hat{E}_{S, j^{(t)}} \geq \sum_{S \in M : S \subseteq S^{(t)}} \hat{P}_S$ and $\left(\sum_{S \in M : S \subseteq S^{(t)}} \hat{P}_S\right) u_{j^{(t)}}^{(t)} - \sum_{S \in M : S \subseteq S^{(t)}} \hat{E}_{S, j^{(t)}} < -\beta/2$. Therefore,

$$
\mathbb{E}\left[\left\|z^{(t)} - y\right\|_2^2 - \left\|u^{(t)} - y\right\|_2^2 \middle| R\left(f\left(x\right)\right) \in S^{(t)}\right] \mathbb{P}\left[R\left(f\left(x\right)\right) \in S^{(t)}\right]
$$

$$
= \left(1 - u_{j^{(t)}}^{(t)}\right) \left(\left(1 + u_{j^{(t)}}^{(t)}\right) \mathbb{P}\left[R\left(f\left(x\right)\right) \in S^{(t)}\right] - 2\mathbb{E}\left[y_{j^{(t)}} \mathbb{I}\left[R\left(f\left(x\right)\right) \in S^{(t)}\right]\right]\right).
$$

We analyze the two factors separately. Since the error associated with bin $S^{(t)}$ and coordinate $j^{(t)}$ is large, we have that

$$
\left(\sum_{S \in M : S \subseteq S^{(t)}} \hat{P}_S\right) u_{j^{(t)}}^{(t)}
$$

$$
< \sum_{S \in M : S \subseteq S^{(t)}} \hat{E}_{S, j^{(t)}} - \frac{\beta}{2}
$$

$$
< \mathbb{E}\left[y_{j^{(t)}} \mathbb{I}\left[R\left(f\left(x\right)\right) \in S^{(t)}\right]\right] + \frac{\beta}{36(\lfloor \log_2 |B| \rfloor + 1)} \left|\left\{S \in M : S \subseteq S^{(t)}\right\}\right| - \frac{\beta}{2}
$$

$$
\leq \mathbb{P}\left[R\left(f\left(x\right)\right) \in S^{(t)}\right] - \frac{17\beta}{36}.
$$

Furthermore, we have a lower on the estimated probability of $S^{(t)}$ $\sum_{S \in M : S \subseteq S^{(t)}} \hat{P}_S \geq \mathbb{P}\left[R\left(f\left(x\right)\right) \in S^{(t)}\right] - \frac{\beta}{36(\lfloor \log_2 |B| \rfloor + 1)} \left|\left\{S \in M : S \subseteq S^{(t)}\right\}\right| \geq \frac{\beta}{6} - \frac{\beta}{12} - \frac{\beta}{36} > 0$ because $S^{(t)} \in G$, which implies that it contains bins from set $B$.

Combining the two inequalities above, we obtain that

$$
1 - u_{j^{(t)}}^{(t)} > 1 - \frac{\mathbb{P}\left[R\left(f\left(x\right)\right) \in S^{(t)}\right] - 17\beta/36}{\mathbb{P}\left[R\left(f\left(x\right)\right) \in S^{(t)}\right] - \beta/36}
$$

$$
= \frac{\beta/2 - \beta/18}{\mathbb{P}\left[R\left(f\left(x\right)\right) \in S^{(t)}\right] - \beta/36} > \frac{4\beta}{9}.
$$

We now bound the second factor.

$$\left(1 + u_{j^{(t)}}^{(t)}\right) \mathbb{P}\left[R\left(f\left(x\right)\right) \in S^{(t)}\right] - 2\mathbb{E}\left[y_{j^{(t)}} \mathbb{I}\left[R\left(f\left(x\right)\right) \in S^{(t)}\right]\right]$$

$$\leq \left(1 + u_{j^{(t)}}^{(t)}\right) \left(\sum_{S \in M: S \subseteq S^{(t)}} \hat{P}_S + \frac{\beta}{36(\lfloor \log_2 |B| \rfloor + 1)} \left|\left\{S \in M : S \subseteq S^{(t)}\right\}\right|\right)$$

$$- 2\left(\sum_{S \in M: S \subseteq S^{(t)}} \hat{E}_{S,j^{(t)}} - \frac{\beta}{36(\lfloor \log_2 |B| \rfloor + 1)} \left|\left\{S \in M : S \subseteq S^{(t)}\right\}\right|\right)$$

$$\leq u_{j^{(t)}}^{(t)} \left(\sum_{S \in M: S \subseteq S^{(t)}} \hat{P}_S\right) - \sum_{S \in M: S \subseteq S^{(t)}} \hat{E}_{S,j^{(t)}} + \sum_{S \in M: S \subseteq S^{(t)}} \hat{P}_S - \sum_{S \in M: S \subseteq S^{(t)}} \hat{E}_{S,j^{(t)}} + \frac{\beta}{9}$$

$$< -\frac{7\beta}{18}.$$

Multiplying the two factors, we see that

$$\mathbb{E}\left[\left\|z^{(t)} - y\right\|_2^2 - \left\|u^{(t)} - y\right\|_2^2 \Big| R\left(f\left(x\right)\right) \in S^{(t)}\right] \mathbb{P}\left[R\left(f\left(x\right)\right) \in S^{(t)}\right] < -\frac{14\beta^2}{81}.$$

At a high level, we have shown that the expected difference in squared error is strictly negative in this case.

Case 2: $z_{j^{(t)}}^{(t)} = \left(\sum_{S \in M: S \subseteq S^{(t)}} \hat{E}_{S,j^{(t)}}\right) / \left(\sum_{S \in M: S \subseteq S^{(t)}} \hat{P}_S\right) \leq 1$. We consider two subcases based on the behavior of $u_{j^{(t)}}^{(t)}$.

Subcase 1: $\sum_{S \in M: S \subseteq S^{(t)}} \hat{E}_{S,j^{(t)}} - \left(\sum_{S \in M: S \subseteq S^{(t)}} \hat{P}_S\right) u_{j^{(t)}}^{(t)} > \beta/2$. Then, it follows that

$$z_{j^{(t)}}^{(t)} - u_{j^{(t)}}^{(t)} = \frac{\sum_{S \in M: S \subseteq S^{(t)}} \hat{E}_{S,j^{(t)}}}{\sum_{S \in M: S \subseteq S^{(t)}} \hat{P}_S} - u_{j^{(t)}}^{(t)} > \frac{\beta}{2\left(\sum_{S \in M: S \subseteq S^{(t)}} \hat{P}_S\right)}$$

and

$$\left(z_{j^{(t)}}^{(t)} + u_{j^{(t)}}^{(t)}\right) \mathbb{P}\left[R\left(f\left(x\right)\right) \in S^{(t)}\right] - 2\mathbb{E}\left[y_{j^{(t)}} \mathbb{I}\left[R\left(f\left(x\right)\right) \in S^{(t)}\right]\right]$$

$$= \left(\frac{\sum_{S \in M: S \subseteq S^{(t)}} \hat{E}_{S,j^{(t)}}}{\sum_{S \in M: S \subseteq S^{(t)}} \hat{P}_S} + u_{j^{(t)}}^{(t)}\right) \mathbb{P}\left[R\left(f\left(x\right)\right) \in S^{(t)}\right] - 2\mathbb{E}\left[y_{j^{(t)}} \mathbb{I}\left[R\left(f\left(x\right)\right) \in S^{(t)}\right]\right]$$

$$< \left(2\frac{\sum_{S \in M: S \subseteq S^{(t)}} \hat{E}_{S,j^{(t)}}}{\sum_{S \in M: S \subseteq S^{(t)}} \hat{P}_S} - \frac{\beta}{2\sum_{S \in M: S \subseteq S^{(t)}} \hat{P}_S}\right) \mathbb{P}\left[R\left(f\left(x\right)\right) \in S^{(t)}\right] - 2\mathbb{E}\left[y_{j^{(t)}} \mathbb{I}\left[R\left(f\left(x\right)\right) \in S^{(t)}\right]\right]$$

$$\leq \left(2\frac{\sum_{S \in M: S \subseteq S^{(t)}} \hat{E}_{S,j^{(t)}}}{\sum_{S \in M: S \subseteq S^{(t)}} \hat{P}_S} - \frac{\beta}{2\sum_{S \in M: S \subseteq S^{(t)}} \hat{P}_S}\right)$$

$$\cdot \left(\sum_{S \in M: S \subseteq S^{(t)}} \hat{P}_S + \frac{\beta}{36(\lfloor \log_2 |B| \rfloor + 1)} \left|\left\{S \in M : S \subseteq S^{(t)}\right\}\right|\right)$$

$$- 2\left(\sum_{S \in M: S \subseteq S^{(t)}} \hat{E}_{S,j^{(t)}} - \frac{\beta}{36(\lfloor \log_2 |B| \rfloor + 1)} \left|\left\{S \in M : S \subseteq S^{(t)}\right\}\right|\right)$$

$$\leq -\frac{\beta}{2} - \frac{\beta^2}{2 \cdot 36(\lfloor \log_2 |B| \rfloor + 1) \sum_{S \in M: S \subseteq S^{(t)}} \hat{P}_S} \left|\left\{S \in M : S \subseteq S^{(t)}\right\}\right| + \frac{\beta}{18} < -\frac{4\beta}{9}.$$

Subcase 2: $\sum_{S\in M:S\subseteq S^{(t)}} \hat{E}_{S,j^{(t)}} - \left(\sum_{S\in M:S\subseteq S^{(t)}} \hat{P}_S\right) u_{j^{(t)}}^{(t)} < -\beta/2$. Then, it follows that

$$z_{j^{(t)}}^{(t)} - u_{j^{(t)}}^{(t)} = \frac{\sum_{S\in M:S\subseteq S^{(t)}} \hat{E}_{S,j^{(t)}}}{\sum_{S\in M:S\subseteq S^{(t)}} \hat{P}_S} - u_{j^{(t)}}^{(t)} < -\frac{\beta}{2\left(\sum_{S\in M:S\subseteq S^{(t)}} \hat{P}_S\right)}$$

and

$$\left(z_{j^{(t)}}^{(t)} + u_{j^{(t)}}^{(t)}\right) \mathbb{P}\left[R\left(f\left(x\right)\right) \in S^{(t)}\right] - 2\mathbb{E}\left[y_{j^{(t)}}\mathbb{I}\left[R\left(f\left(x\right)\right) \in S^{(t)}\right]\right]$$

$$= \left(\frac{\sum_{S\in M:S\subseteq S^{(t)}} \hat{E}_{S,j^{(t)}}}{\sum_{S\in M:S\subseteq S^{(t)}} \hat{P}_S} + u_{j^{(t)}}^{(t)}\right) \mathbb{P}\left[R\left(f\left(x\right)\right) \in S^{(t)}\right] - 2\mathbb{E}\left[y_{j^{(t)}}\mathbb{I}\left[R\left(f\left(x\right)\right) \in S^{(t)}\right]\right]$$

$$> \left(2\frac{\sum_{S\in M:S\subseteq S^{(t)}} \hat{E}_{S,j^{(t)}}}{\sum_{S\in M:S\subseteq S^{(t)}} \hat{P}_S} + \frac{\beta}{2\sum_{S\in M:S\subseteq S^{(t)}} \hat{P}_S}\right)$$

$$\cdot \left(\sum_{S\in M:S\subseteq S^{(t)}} \hat{P}_S - \frac{\beta}{36(\lfloor\log_2|B|\rfloor + 1)}\left|\left\{S \in M : S \subseteq S^{(t)}\right\}\right|\right)$$

$$- 2\left(\sum_{S\in M:S\subseteq S^{(t)}} \hat{E}_{S,j^{(t)}} + \frac{\beta}{36(\lfloor\log_2|B|\rfloor + 1)}\left|\left\{S \in M : S \subseteq S^{(t)}\right\}\right|\right)$$

$$\geq \frac{\beta}{2} - \frac{\beta^2}{2\cdot 36(\lfloor\log_2|B|\rfloor + 1)\sum_{S\in M:S\subseteq S^{(t)}} \hat{P}_S}\left|\left\{S \in M : S \subseteq S^{(t)}\right\}\right| - \frac{\beta}{18} > \frac{4\beta}{9}.$$

Therefore, in both subcases the expected difference in squared error is also strictly negative. Specifically, we have

$$\mathbb{E}\left[\left\|z^{(t)} - y\right\|_2^2 - \left\|u^{(t)} - y\right\|_2^2 \,\bigg|\, R\left(f\left(x\right)\right) \in S^{(t)}\right] \mathbb{P}\left[R\left(f\left(x\right)\right) \in S^{(t)}\right]$$

$$< -\left(\frac{4\beta}{9}\right)\frac{\beta}{2\left(\sum_{S\in M:S\subseteq S^{(t)}} \hat{P}_S\right)}$$

$$< -\frac{\beta^2}{9}.$$

because $\sum_{S\in M:S\subseteq S^{(t)}} \hat{P}_S \leq \mathbb{P}\left[R\left(f\left(x\right)\right) \in S^{(t)}\right] + \frac{\beta}{36} \leq 2$.

We notice that in both cases

$$\mathbb{E}\left[\left\|z^{(t)} - y\right\|_2^2 - \left\|u^{(t)} - y\right\|_2^2 \,\bigg|\, R\left(f\left(x\right)\right) \in S^{(t)}\right] \mathbb{P}\left[R\left(f\left(x\right)\right) \in S^{(t)}\right] < -\frac{\beta^2}{9}.$$

$\square$

**Lemma 23** (Lemma 13 restated). *The squared error at time step $0$ is* $\mathbb{E}\left[\|h_0(x) - y\|_2^2\right] \leq \mathbb{E}\left[\|f\left(x\right) - y\|_2^2\right] + O\left(\beta\right)$.

*Proof.* By the definition of $\rho$, $h_0(x) = \rho\left(R\left(f\left(x\right)\right)\right)$ and $f(x)$ correspond to the same level set when they get rounded by $R$. Therefore, they are at most $1/\lambda$ apart in every coordinate. Additionally, the coordinates of $f(x)$ and $h_0(x)$ add up to $1$. Since $y$ is the one-hot encoding of a label, we obtain

that

$$\|h_0(x) - y\|_2^2$$

$$= \|h_0(x) - y\|_2^2 - \|f(x) - y\|_2^2 + \|f(x) - y\|_2^2$$

$$\leq \|h_0(x)\|_2^2 - \|f(x)\|_2^2 + 2\max_{j \in [k]} |h_0(x)_j - f(x)_j| + \|f(x) - y\|_2^2$$

$$\leq \left( \max_{j \in [k]} |h_0(x)_j - f(x)_j| \right) \sum_{j \in [k]} (|h_0(x)_j| + |f(x)_j|) + 2\max_{j \in [k]} |h_0(x)_j - f(x)_j| + \|f(x) - y\|_2^2$$

$$\leq \frac{1}{\lambda} \cdot 4 + \|f(x) - y\|_2^2 = \frac{4}{\lceil 1/\beta \rceil} + \|f(x) - y\|_2^2.$$

$\square$

### A.4 PROOFS FROM SUBSECTION 3.3

**Lemma 24** (Lemma 14 restated). *Assuming that $A_1$, $A_2$ and $A_3$ hold, the algorithm terminates after at most $O\left(1/\beta^2\right)$ time steps.*

*Proof.* Assuming that events $A_1, A_2$ and $A_3$ hold, we apply Lemmata 11 and 12 to obtain the following bound

$$\mathbb{E}\left[\|h(x) - y\|_2^2\right] - \mathbb{E}\left[\|\rho\left(R\left(f\left(x\right)\right)\right) - y\|_2^2\right] \leq -\frac{\beta^2}{9}T + \frac{4}{\lceil 1/\beta \rceil} \log_2\left(\frac{36}{\beta}\right).$$

Moreover, since the squared loss is always bounded between 0 and 1 we have

$$-1 \leq -\frac{\beta^2}{9}T + \frac{4}{\lceil 1/\beta \rceil} \log_2\left(\frac{36}{\beta}\right)$$

which implies that the algorithm must terminate after

$$T \leq \frac{9 + \frac{36}{\lceil 1/\beta \rceil} \log_2\left(\frac{36}{\beta}\right)}{\beta^2}$$

time steps. $\square$

**Lemma 25** (Lemma 15 restated). *Assuming that $A_1$, $A_2$ and $A_3$ hold, the $\ell_p$ calibration error $(Err_p(h))^p$ is bounded by $O(\beta^{p-1})$.*

*Proof.* Let $T$ be the time step when the algorithm terminates. We analyze the error under the assumption that $A_1, A_2$ and $A_3$ hold. We show that for all $v \in V_\lambda^k$ and all $j \in [k]$, $\text{Err}(h, v, j) \leq \beta$.

A point $x$ gets a prediction $h(x)$ that gets rounded to level set $v$ in one of two ways:

1. if $v$ is not a high-probability bin, then the initial prediction $f(x)$ gets rounded to $v$, or

2. if there exists a group of bins $S \in G$ such that $R\left(\text{pred}(S)\right) = v$, then the initial prediction $f(x)$ is in a high-probability bin that, through the calibration algorithm gets mapped to group $S$.

Note that both cases can be true simultaneously for a fixed $v$. In the second case, due to the termination criterion of the algorithm, $\forall j \in [k]$,

$$\hat{\text{Err}}(S, j) = \left| \left( \sum_{S' \in M: S' \subseteq S} \hat{P}_{S'} \right) \text{pred}\left(S\right)_j - \sum_{S' \in M: S' \subseteq S} \hat{E}_{S', j} \right| \leq \frac{\beta}{2}.$$

For the true error of $v \in V_\lambda^k$ and $j \in [k]$, we have that

$\mathrm{Err}(h, v, j)$

$= \left| \mathbb{E}_{(x,y) \sim D} \left[ (h(x)_j - y_j) \, \mathbb{I} \left[ R\left( h\left( x \right) \right) = v \right] \right] \right|$

$\leq \left| \mathbb{E}_{(x,y) \sim D} \left[ (h(x)_j - y_j) \, \mathbb{I} \left[ R\left( h\left( x \right) \right) = v \text{ and } R(f(x)) \in B \right] \right] \right|$

$\quad + \left| \mathbb{E}_{(x,y) \sim D} \left[ (h(x)_j - y_j) \, \mathbb{I} \left[ R\left( h\left( x \right) \right) = v \text{ and } R(f(x)) \notin B \right] \right] \right|$

$\leq \left| \mathbb{P} \left[ R(f(x)) \in S \right] \cdot \mathrm{pred}(S)_j - \mathbb{E}_{(x,y) \sim D} \left[ y_j \mathbb{I} \left[ R(f(x)) \in S \right] \right] \right| \cdot$

$\mathbb{I} \left[ \exists S \in G : R\left( \mathrm{pred}(S) \right) = v \right] + \mathbb{P} \left[ R(f(x)) = v \right] \mathbb{I} \left[ v \notin B \right]$

$\leq ( |( \sum_{S' \in M : S' \subseteq S} \hat{P}_{S'} ) \mathrm{pred}(S)_j - \sum_{S' \in M : S' \subseteq S} \hat{E}_{S',j}|$

$+ \dfrac{2\beta}{36(\lfloor \log_2 |B| \rfloor + 1)} |\{ S' \in M : S' \subseteq S \}|) \mathbb{I} \left[ \exists S \in G : R\left( \mathrm{pred}(S) \right) = v \right] + \left( \dfrac{\beta}{6} + \dfrac{\beta}{12} \right) \mathbb{I}[v \notin B]$

$\leq \left( \dfrac{\beta}{2} + \dfrac{\beta}{18} \right) \mathbb{I} \left[ \exists S \in G : R\left( \mathrm{pred}(S) \right) = v \right] + \dfrac{\beta}{4} \mathbb{I}[v \notin B]$

$\leq \beta$

Therefore,

$$\sum_{v \in V_\lambda^k} \sum_{j=1}^k (\mathrm{Err}(h, v, j))^p$$

$$\leq ( \sum_{v \in V_\lambda^k} \sum_{j=1}^k \mathrm{Err}(h, v, j)) \max_{v \in V_\lambda^k, j \in [k]} (\mathrm{Err}(h, v, j))^{p-1}$$

$$\leq ( \sum_{v \in V_\lambda^k} \sum_{j=1}^k \left( \mathbb{E}_{(x,y) \sim D} \left[ h(x)_j \mid R(h(x)) = v \right] \right.$$

$$+ \mathbb{E}_{(x,y) \sim D} \left[ y_j \mid R(h(x)) = v \right] ) \, \mathbb{P} \left[ R(h(x)) = v \right] ) \beta^{p-1}$$

$$\leq 2\beta^{p-1}$$

This holds because for all $v \in V_\lambda^k$, $\sum_{j=1}^k \mathbb{E}_{(x,y) \sim D} \left[ h(x)_j \mid R(h(x)) = v \right] = 1$. As a result we get that $\mathbb{P} \left[ \mathrm{Err}_p(h) > \left( 2\beta^{p-1} \right)^{1/p} \mid A_1, A_2, A_3 \right] = 0$. $\qquad \square$

**Lemma 26** (Lemma 16 restated). *Assuming that $A_1$, $A_2$ and $A_3$ hold, the algorithm terminates in time $O\left( \frac{k}{\beta^2} \log^3 \left( \frac{1}{\beta} \right) \log \left( \frac{k}{\beta\delta} \right) \right)$.*

*Proof.* Assuming that $A_1$, $A_2$ and $A_3$ hold, Algorithm 2 has time complexity $O\left( \mathrm{poly} \left( \frac{1}{\beta}, k \right) \right)$, where poly denotes a polynomial function. We analyze the time complexity of each phase of the algorithm.

Phase 1: Identifying high-probability bins. This phase requires $O(n)$ time, where $n$ is the number of samples used to estimate $\hat{\mu}_v$. According to the analysis in Subsection 3.1, $n = O\left( \frac{1}{\beta^2} \log \left( \frac{k}{\beta\delta} \right) \right)$. Notably, this step avoids iterating over all bins in $V_\lambda^k$ by examining only bins containing input samples. This can be efficiently implemented using a dictionary/hash table where keys represent bins and values are lists of samples in each bin. The dictionary size equals the number of non-empty bins. From this point forward the algorithm operates exclusively on the high probability bins in $B$, whose cardinality is linear in $\frac{1}{\beta}$.

Phase 2: Initializing data structures $M$ and $G$. The initialization requires time linear in $|B|k = O\left( \frac{1}{\beta} k \right)$. For the computation of the error, the algorithm first estimates $\hat{P}$ and $\hat{E}$. Similarly to Phase 1, this part requires $O(mk)$ time, where $m$ is the number of samples used to estimate $\hat{P}$ and

$\hat{E}$. By the analysis in Subsection 3.1, the number of these samples is $O\left(\frac{1}{\beta^2}\log^3\left(\frac{1}{\beta}\right)\log\left(\frac{k}{\beta\delta}\right)\right)$. Then, the algorithm projects every vector in $G$ using $\rho$ to get the values of pred, which takes time $O(k)$. More specifically, $r(v)$ is of the form $r(v)_i = v_i + z$, where $z = \frac{1-\sum_{i\in[k]} v_i}{k}$. Finally, the computation of the estimated errors takes $O(k|B|) = O\left(\frac{k}{\beta}\right)$ time.

Phase 3: Calibration. The algorithm calibrates predictions for bins in $B$ by executing at most $O\left(\frac{1}{\beta^2}\right)$ iterations. Each iteration performs a polynomial number of operations in $k$ and $\frac{1}{\beta}$. More specifically, searching in $G$ for the large-error group can take at most $O(\log(|B|k))$ time if we store the errors of the groups in $G$ in a priority queue. The computation of $z^{(t)}$ takes time at most $O(|S^{(t)}| + k)$. By Lemma 10 we know that $|S^{(t)}| = O(\log|B|)$. After the algorithm computes $z^{(t)}$, it projects it to the simplex using $\pi$, which can be done in time $O(k\log(k))$. The search for groups to merge can be implemented using a hash table whose keys are $R(\text{pred}(S))$ for $S$ in $G$ and values are the groups corresponding to each key and, hence, takes constant time. The total number of merges in $G$ and $M$ throughout the entire algorithm is bounded by $|B|$, since we begin with $|B|$ groups and only merge. Therefore, the parts of the algorithm that perform the merges get executed at most $O(\frac{1}{\beta})$ times in total. Merging two groups in $G$ takes time $O(|B|k)$ since we only update the predictions for the affected bins. The merge in $M$ takes time $O(k)$ since we only adjust the estimates for $S_1$ and $S_2$. The error computation step runs in time linear in $k\log|B|$ since by Lemma 10 the sum used to estimate the probability of $S^{(t)}$ consists of at most $O(\log|B|)$ terms.

Combining the analyses of the three phases, we conclude that the algorithm's time complexity is $O\left(\frac{k}{\beta^2}\log^3\left(\frac{1}{\beta}\right)\log\left(\frac{k}{\beta\delta}\right)\right)$. □

USE OF LLMS

We used Claude Opus 4.1 by Anthropic for grammar and spell checking.