# OpenReview forum: "Sample-efficient Multiclass Calibration under $\ell_{p}$ Error"
_ICLR.cc/2026/Conference — Submitted to ICLR 2026_

### Official Review · Reviewer_9XGc · 2025-10-31

**Soundness:** 4
**Presentation:** 3
**Contribution:** 3
**Rating:** 4
**Confidence:** 3

**Summary:**

The paper is theoretically focused. It introduces a new definition of multiclass calibration error that unifies and interpolates between existing notions. Additionally, it provides an algorithm that achieves polynomial sample complexity, which is near-optimal in some cases.

The main body of the paper presents key theorems, lemmas, and proofs, while detailed derivations are deferred to the appendix. The work also includes a discussion of related literature in calibration and adaptive data analysis, highlighting how the proposed framework unifies and extends existing results. There are no experiments or empirical illustrations, as the contribution is mainly theoretical.

**Strengths:**

The paper is original in formulating the new $\ell_p$ calibration error, which unifies and interpolates between existing calibration notions while removing the exponential dependence on the number of classes found in prior work. The theoretical analysis appears rigorous and internally consistent, introducing an interesting adaptive-data-analysis argument that achieves near-optimal sample complexity. The paper is well structured for a theoretical contribution. Overall, it seems to provide a meaningful and technically solid advance in the theory of multiclass calibration, but I did not verify the proofs.

**Weaknesses:**

1. While the theoretical contribution appears strong, the work does not include an empirical analysis that could help clarify the behavior of the proposed algorithm. The practicality of the method is not discussed; although the algorithm is polynomial, the constants may be large, and it is unclear how it would scale in realistic settings. It is not clear how tight the upper bounds are in practice, this could have been investigated empirically on synthetic datasets.

2. The paper is highly technical and difficult to follow:
* There are no illustrative diagrams or figures that could help explain the algorithm. For example, for 3 classes it would be possible to visualise the simplex, the initial discretization corresponding to $\lambda$, and the following merging steps in G and M. In particular, I would have appreciated more intuitive descriptions about the interplay of G and M, both in the general case and for some concrete toy example with 3 classes.
* The three events are introduced without an explanation about why these particular events are good to consider.
* I think that pushing some lemmas to the appendix and making more space for illustrations and empirical analysis would have made sense.

3. Some minor weaknesses are reflected in the questions below.

**Questions:**

1. In the introduction (lines 25–28), could the authors clarify how “trustworthiness” and “interpretability” relate to calibration? The connection between these ideas could be made more explicit.
2. Line 103 mentions that “for interpretability reasons, the outputs of our calibrated predictor should be probability distributions.” Could the authors elaborate on this? Shouldn’t the predictor’s outputs be probability distributions regardless of interpretability concerns, e.g. for automatic cost-sensitive decision-making purposes?
3. It might be better to define the asymptotic notation with a tilde ($\tilde{O}$) explicitly, as it appears in Theorems 1 and 6 but is never formally introduced.
4. How practical would it be to implement the proposed algorithm for different values of $k$ and $\varepsilon$? It would be helpful if the authors could provide an estimate or discussion of the expected runtime or sample requirements in practical settings to better understand whether the method could be applied in practice or remains primarily of theoretical interest.
5. Minor (Line 621): There are missing spaces in the proof of Lemma 2, which slightly affects readability.

**Details Of Ethics Concerns:**

-

---

> ### Author Response · Authors · 2025-11-18
> **Response to Reviewer 9XGc**
>
> Thank you for your valuable comments and suggestions. We address your specific questions below.
>
> **Example for $V_{\lambda}^k$, $G$ and $M$**: For 3 classes (k=3), $p = \infty$ and $\epsilon=0.5$, we have $\beta = \epsilon = 0.5$ and $\lambda = 2$. Therefore, $V_2^3 = \lbrace(0,0,0), (0.5, 0, 0), (0, 0.5, 0), (0, 0, 0.5), (0,0,1), (0,1,0), (1,0,0), (0,0.5,0.5), (0.5, 0, 0.5), (0.5, 0.5,0)\rbrace$.
> Now, consider an instance of the problem where the large bins are $B = \lbrace(0,0,1), (0,0.5,0.5), (0.5, 0,0.5)\rbrace$.
> Then, $G$ and $M$ are initialized as $\lbrace\lbrace(0,0,1)\rbrace, \lbrace(0,0.5,0.5)\rbrace, \lbrace(0.5, 0,0.5)\rbrace\rbrace$.
> Suppose the estimates indicate that all three should get the same prediction and the correction order is $v_1 = (0,0,1)$, $v_2 = (0,0.5,0.5)$ and $v_3=(0.5,0,0.5)$. A potential progression of $M$ and $G$ is:
>
> - Step 1:
> $G = M = \lbrace\lbrace(0,0,1)\rbrace, \lbrace(0,0.5,0.5)\rbrace, \lbrace(0.5, 0,0.5)\rbrace\rbrace$
> - Step 2:
> $G = M = \lbrace\lbrace(0,0,1), (0,0.5,0.5)\rbrace, \lbrace(0.5, 0,0.5)\rbrace\rbrace$
> - Step 3:
> $G = \lbrace\lbrace(0,0,1), (0,0.5,0.5), (0.5, 0,0.5)\rbrace\rbrace$
> $M = \lbrace\lbrace(0,0,1), (0,0.5,0.5)\rbrace, \lbrace(0.5, 0,0.5)\rbrace\rbrace$
>
> **“The three events…”**: For the algorithm to operate correctly, the empirical estimates are required to be close to the true values. The three events formalize this requirement. In particular, event $A_1$ ensures that the estimates used to build the high-probability bins $B$ are accurate, guaranteeing that $B$ contains bins with large probability masses. Event $A_2$ ensures that probability estimates for groups in $M$ are close to their true values, while event $A_3$ ensures that mean label estimates are accurate. Together, these events enable the algorithm to correctly adjust predictions and merge bins as needed.
>
> **Question 1**: We refer the reviewer to the discussion in the introduction of the seminal paper by Guo et al. (2017). Calibration enables models to provide reliable confidence estimates alongside their predictions. When a model’s predictions accurately reflect the true likelihood of events, users can make informed decisions about when to rely on the model’s output versus seeking alternative prediction approaches. This capability is valuable in cost-sensitive applications where errors carry significant consequences. Additionally, calibrated predictors can be interpreted as probability distributions, facilitating their integration into larger models and downstream decision-making frameworks.
>
> **Question 2**: We agree with the reviewer that predictors used for downstream decision-making should output probability distributions, which is why our algorithm guarantees this property. We emphasize this in the paper because some previous algorithms lacked this guarantee.
>
> **Question 3**: We include this in the posted revision.
>
> **Question 4**: In the revision, we include the detailed analysis of the time complexity of the algorithm. We show that a naive implementation terminates in time $O\left(\frac{k}{\beta^2} \log^3\left(\frac{1}{\beta}\right)\log\left(\frac{k}{\beta\delta}\right)\right)$, where $\beta = \varepsilon^{p/(p-1)} 2^{-1/(p-1)}$.

---

### Official Review · Reviewer_StCW · 2025-10-31

**Soundness:** 1
**Presentation:** 1
**Contribution:** 2
**Rating:** 2
**Confidence:** 2

**Summary:**

This paper studies efficient calibration of existing predictors while controlling the accuracy loss. It generalizes (interpolate) the existing sample error definitions of $\ell_1$ and $\ell_\infty$ to $\ell_p$ and preserves polynomial sample complexity.

**Strengths:**

Originality:
The use of adaptive data analysis can be considered creative.

Quality:
Some intermediate results are technically correct.

Clarity:
General findings are understandable.

Significance:
The theoretical novel findings are the extension of the existing calibration error definitions in the form $\ell_p$ to $p\in(1,\infty)$ and applying suitable calibrations with efficient sampling.

**Weaknesses:**

I argue for rejection. Below is my reason.

I understood the general direction of the paper. However, most of the details are not apparent to me. The paper has major clarity issues. I am unsure about the meaning and definitions of many of the notations, so it is extremely difficult for me to verify (even coarsely) many of the intermediary claims and understand the explanations in between. For example, when you say $O(\log(|B|))$ collections of bins, how are these collections separated? Possibly, the paper is too condensed. Even so, I believe the paper needs to be revised at least to become more reader-friendly.

**Questions:**

Questions:

Page 1 Line 53: weaker definition is unclear. What is it exactly?

Page 2 Line 84: why $1/\epsilon^2$?

Page 3 Line 112: intuition regarding $1/\epsilon^3$ is unclear. Can you explain more?

Page 5 Line 266: why do they never split?

Page 6 Line 323: using $O(...)$ what exactly?

Page 8 Line 379: why are the sizes powers of $2$?


Suggestions:

Page 5 Definition 5: the error definition seems unconventional since the probabilities are also taken to the power of $p$. The effect of this in the final findings needs to be addressed.

Page 6 Line 290: this paragraph is too convoluted and does not help digestion.

---

> ### Author Response · Authors · 2025-11-18
> **Response to Reviewer StCW**
>
> Thank you for your detailed review and suggestions. We address your questions below.
>
> “**[When] you say $O(\log(|B|)$ collections of bins, how are these collections separated?**”: In line 301 we explain that we partition $M$ (which consists of groups of bins) into collections of groups of bins of the same size. Since we have at most $O(log|B|)$ different group sizes  (all group sizes are powers of $2$), and we have one collection per size, we maintain $O(\log|B|)$ many collections, whose union contains all the bins in $M$.
>
> **Question 1**: In our notation, the calibration definition from Haghtalab et al. (2023) is $\max_{v\in V_{\lambda}^k, j \in [k]}|\mathbb{E}[f(x)_j-y_j|R(f(x))=v]|\mathbb{P}[R(f(x))=v]$.
>
> **Question 2**: Imagine we have two Bernoulli distributions with probabilities $p_1 =\frac{1}{2}$ and $p_2 =\frac{1}{2}+\epsilon$ and we want to understand which distribution $n$ iid samples $\lbrace X_i \rbrace_{i\in[n]}$ are drawn from. Let $D_i$ be the distribution of the entire sequence of $n$ samples from the Bernoulli distribution with probability $p_i$. We can only tell the two distributions $D_1$ and $D_2$ apart if their statistical distance is at least, say, $\frac{1}{2}$. This implies their KL-divergence is at least $\frac{1}{2}$ by Pinsker’s inequality. The KL divergence between 1 Bernoulli sample with probability $p_1$ and 1 Bernoulli sample with probability $p_2$ is $\Theta(\epsilon^2)$. Since the $n$ samples are independent, we can add the KL divergences (chain rule of KL divergence) and the KL-divergence between $D_1$ and $D_2$ is $n \cdot \Theta(\epsilon^2)$. The KL-divergence is at least $\frac{1}{2}$ only if $n * \Theta(\epsilon^2) \ge \frac{1}{2}$. This simplifies to $n=\Omega(1/\epsilon^2)$. The reviewer can also consult lecture notes on hypothesis testing such as https://www.cs.cornell.edu/courses/cs6781/2020sp/lectures/15_sequential-notes.pdf
>
> **Question 3**: There are $\exp(\frac{1}{\epsilon})$ many subsets we need to estimate, and for each subset, we face the same question as distinguishing $D_1$ and $D_2$ above. If we want to be correct on all of them, then we can only allow the failure probability for each subset to be $\frac{1}{\exp(1/\epsilon)}$. The bound above can be strengthened to $\frac{\log(1/\text{failure probability})}{\epsilon^2}$ as described in the lecture notes above (using a strengthened version of Pinsker’s such as Bretagnolle-Huber’s). Substituting the failure probability = \frac{1}{\exp(1/\epsilon)} gives the \frac{1}{\epsilon^3} bound.
>
> **Question 4**: In Algorithm 1, the sole operation applied to groups of bins in $G$ is merging, executed in line 281.
>
> **Question 5**: We mean $O(\ldots)$ samples and have corrected this in the revision.
>
> **Question 6**: $M$ contains only singleton groups when initialized in line 332 of Algorithm 2. Subsequently, the algorithm merges groups of equal size in $M$ ( lines 355, 356). We can prove by induction that the sizes of all the groups in $M$ are powers of $2$. *Base case*: All groups initially have size $1=2^0$.
> *Inductive hypothesis*: After a step $t$, all groups in $M$ have sizes that are powers of $2$.
> *Inductive step*: At time step $t+1$, suppose without loss of generality that the algorithm merges two groups $S_1$ and $S_2$ of sizes $2^{i_1}$ and $2^{i_2}$ respectively. Since the algorithm only merges groups of equal size, we have $i_1=i_2$. Their union has size $2^{i_1}+2^{i_2} = 2 * 2^{i_1} = 2^{i_1+1}$, which is also a power of $2$.

---

### Official Review · Reviewer_vANQ · 2025-10-31

**Soundness:** 3
**Presentation:** 2
**Contribution:** 3
**Rating:** 4
**Confidence:** 3

**Summary:**

This paper introduces a new Lp calibration error definition for multiclass classification that interpolates between expected calibration error (p=1) and maximum calibration error (p=∞). The authors present an algorithm that calibrates any given k-class predictor using polynomial samples for all p > 1 and achieves near-optimal dependence on the error parameter for p=∞.

**Strengths:**

- The polynomial sample complexity for all p > 1 is a significant achievement
- The near-optimal Õ(1/ε²) bound for p=∞ substantially improves over prior exponential-sample methods.
- The proposed discretization scheme using cleverly reduces the prediction space from exponential to polynomial by enforcing the simplex constraint.
- The algorithm preserves the accuracy of the given predictor (up to small discretization error)

**Weaknesses:**

**Conflicting definition of Lp calibration error**

The paper introduces "Lp calibration error" as a novel contribution, but this term is already established in the literature with a different definition.

- https://arxiv.org/pdf/1909.10155 Eq (1) and paragraph below
- https://proceedings.neurips.cc/paper_files/paper/2022/file/33d6e648ee4fb24acec3a4bbcd4f001e-Paper-Conference.pdf Eq (1)
- https://proceedings.neurips.cc/paper_files/paper/2022/file/3915a87ddac8e8c2f23dbabbcee6eec9-Paper-Conference.pdf Eq (2)

Reusing established terminology for a different concept creates confusion and makes it difficult to compare with related work.

**Complete absence of experiments**

The paper is purely theoretical with zero empirical validation. No experiments on synthetic or real datasets demonstrating:
- Practical sample complexity vs. theoretical bounds
- Computational runtime
- Comparison to existing calibration methods
- Behavior across different values of p
- Quality of calibrated predictions
- Density of theoretical presentation

**The paper presents a very dense sequence of theoretical results**

 While mathematical rigor is commendable, the presentation feels overwhelming and difficult to parse. The paper would benefit from 1) providing more intuitive explanations before formal statements 2)  including proof sketches or key ideas for the most important lemmas in the main text 3) potentially consolidating some lemmas or moving technical details to supplementary material to improve readability

**Limited discussion of p=1 case**

The ECE (p=1) case is mentioned but explicitly excluded from the algorithm. This seems like an important practical case that practitioners care about.

Minor:
- Typo L381: there are can be

**Questions:**

How does the proposed definition relate to established concepts in calibration: top-label, marginal and canonical calibration?

How should practitioners choose between your definition and these alternatives? A formal hierarchy or comparison of calibration definitions would clarify the paper's positioning.

---

> ### Author Response · Authors · 2025-11-18
> **Response to Reviewer vANQ**
>
> Thank you for your valuable feedback and suggestions. We address your questions and comments below.
>
> **"Limited discussion of $p=1$ case"**: We would like to clarify that the $\ell_{1}$ (ECE) error requires exponentially many samples in the number of classes to estimate for multiclass predictors (Gopalan et al. 2024), making calibration with polynomially many samples infeasible for any algorithm, including ours. We discuss this sample complexity barrier explicitly in the revised version.
>
> **“How does the proposed definition relate to established concepts in calibration: top-label, marginal and canonical calibration? How should practitioners choose between your definition and these alternatives?”**: When selecting a (conditional) calibration error metric, two factors should be considered:
>  1. **What events to condition on**: This choice depends on which part of the distribution is meaningful for the application. Top-label calibration is appropriate when we care about the predicted probability of the predicted label being meaningful. However, this approach does not guarantee that the predictions assigned to other classes are calibrated. Canonical calibration is needed when there is a downstream decision task that requires knowledge of the full distribution over labels (such as conformal prediction). This stronger guarantee conditions on the entire prediction vector rather than just the top prediction. Our error metric is of this type, you can see this because for $\ell_p$ error zero, the predictor is perfectly canonically calibrated.
>  2. **How to measure the calibration error**: This refers to the distance metric used to quantify the deviation from perfect calibration given the chosen conditioning events. These metrics can be applied to different kinds of calibration. Several options have been suggested. For example, the Expected Calibration Error (ECE) uses the sum of the absolute differences between the prediction and true probability of the class, multiplied by the probability mass of this group. In practice, calibration is evaluated with respect to available samples, which implicitly ignores low-probability bins. This practical reality naturally aligns with our approach, whereas the previous theoretical definition of $\ell_p$ calibration error does not account for this sample-based constraint.

---

### Official Review · Reviewer_7mSx · 2025-10-31

**Soundness:** 3
**Presentation:** 2
**Contribution:** 3
**Rating:** 6
**Confidence:** 3

**Summary:**

This manuscript addresses the challenge of achieving sample-efficient multiclass calibration, particularly in scenarios where the prediction space grows exponentially with the number of classes $k$. The authors introduce a new definition of calibration error: $\ell_p$ calibration error. This definition is proposed to interpolate between definitions requiring exponential samples and those requiring polynomial samples. The main result Theorem 6 demonstrates an algorithm that can calibrate any given predictor $f$ to achieve low $\ell_p$ calibration error ($\leq \epsilon$) for all $p > 1$, using only a sample complexity that is polynomial in $k$ and $1/\epsilon$. Furthermore, for the case $p=\infty$, the sample complexity achieves a near-optimal dependence on the error parameter $\epsilon$, scaling as $O(1/\epsilon^2)$ up to logarithmic factors. The algorithm is designed to preserve the accuracy of the original predictor $f$, keeping its squared error within an additive term $\tilde{O}(\epsilon^{p/(p-1)} / 2^{1/(p-1)})$. An important technical contribution lies in the analysis of adaptive data analysis for the iterative calibration process which yields an overhead in sample complexity that is only logarithmic in $1/\epsilon$ and improves upon prior polynomial overheads. The algorithm relies on identifying high-probability bins and adaptively merging groups of bins by utilizing a unique discretization scheme that is significantly smaller than previous approaches.

**Strengths:**

1. The $\ell_p$ calibration error definition naturally addresses the issue of the exponential complexity associated with strong error notions in the multiclass setting.

2. The algorithm achieves polynomial sample complexity in the number of classes $k$ for the entire range $p>1$ which provides a powerful tool for multiclass calibration where previous efforts either achieved weak guarantees or required exponential samples.

3. The method successfully calibrates an existing predictor while bounding the resultant decrease in accuracy by a small additive term.

4. The technical result bounding the sample complexity overhead due to adaptive data analysis to $O(\log(1/\epsilon))$ is a significant improvement over standard differential privacy composition techniques that yield polynomial or square-root overheads.

5. The used prediction space partition provides a polynomial bound on the number of bins, contrasting sharply with the exponential size of the space used in prior work.

**Weaknesses:**

1. The work is purely theoretical. While the theoretical contribution seems strong, empirical evidence is required to validate the practical feasibility and tightness of the bounds. This especially true regarding the complexity constants hidden in the $\tilde{O}$ notation and the effect of the adaptive merging process.

2. The calibration procedure involves correcting prediction vectors and projecting them onto the probability simplex. As noted by the authors, this projection can alter previously calibrated coordinates and needs careful selection to preserve accuracy. While the analysis addresses this especially in Lemma 10, 11, 12; the practical stability and computational efficiency of these projection steps might be a concern.

3. The bounds explicitly depend on $p$, particularly the factor $2^{1/(p-1)}$ and $\epsilon^{p/(p-1)}$. When $p$ is close to 1, this factor becomes large. This reflects the increased difficulty when approaching the ECE ($p=1$) definition since the algorithm cannot handle at that endpoint. Further discussions on practical guidance for selecting $p$ would enhance clarity.

**Questions:**

1.  The case $p=1$ corresponds to the expected calibration error (ECE). Could you eloborate on why the algorithm fails for $p=1$? Is there an intrinsic limitation preventing the extension of the current adaptive analysis or error bounding techniques to this critical case? Is there any intuitive reason?

2.  The definition of the $\ell_p$ calibration error incorporates the probability mass of the bin into the $p$-exponent. How does this definition practically compare to other proposed $\ell_p$-style measures in the literature (e.g., Kumar et al., 2019; Popordanoska et al., 2022) in terms of interpretability or robustness beyond the theoretical advantage of testability?

3.  The algorithm's efficiency relies on the adaptive data analysis achieving logarithmic overhead, which leverages the disjointness property of groups in $M$. Does the implementation of the core adaptive mechanism require the strong composition property of differential privacy in practice, or is this primarily an analytical tool to derive the sample complexity bound? Clarifying the practical requirements for achieving this logarithmic overhead would be valuable.

4.  The final predictor $h(x)$ uses the nearest valid probability vector $\rho(R(f(x)))$ if $f(x)$ falls into a low-probability bin (outside $B$). Since $B$ contains bins with estimated probability mass $\mû_v \geq \beta/6$, can the authors quantify how large the calibration error might be for these low-probability bins that are not explicitly modified or tracked by the iterative process?

---

> ### Author Response · Authors · 2025-11-18
> **Response to Reviewer 7mSx**
>
> Thank you for your careful review. We address your questions and comments below.
>
> **Weakness 2**: Without the projection step, the algorithm would be simpler but could produce predictors $h$ whose outputs violate the constraint that probabilities sum to one. As Reviewer 9XGc also points out, having predictors that output probability distributions is crucial for downstream decision making.
>
> **Question 1/Weakness 3**: The $\ell_{1}$ (ECE) error requires exponentially many samples in the number of classes to estimate for multiclass predictors (Gopalan et al. 2024), making calibration with polynomially many samples infeasible for any algorithm, including ours. We discuss this sample complexity barrier explicitly in the revised version.
>
> **Question 2**: In practice, calibration is evaluated with respect to available samples, which implicitly ignores low-probability bins. Intuitively, since the dimension of the prediction vector is the number of classes, the number of bins is exponential in the number of classes. By the pigeonhole principle, either the number of samples is exponential or many bins will have 0 samples. Our approach makes explicit the practical consideration of ignoring low-probability bins, whereas the previous theoretical definition of $\ell_p$ calibration error does not account for this sample-based constraint.
>
> **Question 3**: One approach to adaptive data analysis relies on differential privacy, where the estimation uses empirical averages with differential privacy noise added. While we suspect that the error can be estimated using noiseless empirical averages, the analysis would be considerably more complex. In practice, however, differential privacy noise would not be necessary.
>
> **Question 4**: In the proof of Lemma 14, we show that if the estimates $\hat{\mu}_v$ are accurate (i.e., event $A_1$ holds), then for every low-probability bin $v$, the error satisfies
> $\text{Err}(h,v,j) = |\mathbb{E}[(h(x)_j-y_j)\mathbb{I}[R(h(x))=v]]| \leq |\mathbb{E}[\mathbb{I}[R(h(x))=v]]|= \mathbb{P}[R(h(x))=v]\leq \mu_v+\frac{\beta}{12} \leq \frac{\beta}{4}$.

---

### Comment · Area_Chair_XJ8p · 2025-11-29

Dear Reviewers,

Authors’ kindly tried to address your concerns. If the responses address your concerns please acknowledge that. If not, please express remaining concerns. Thanks for your efforts!

Best, AC

---

### Meta-Review · Area_Chair_XJ8p · 2026-01-03

**Summary:**

The meta-review centers on two critical failures that outweigh the paper's theoretical novelty: a fundamental conflict in terminology and an insufficiently explained phase transition in sample complexity. While the authors propose an algorithm for multiclass calibration with polynomial sample complexity for $p > 1$, the submission reuses established terms ("$L_p$ calibration error") to describe a new metric, creating significant confusion. Furthermore, the paper fails to provide a satisfying intuitive or theoretical explanation for why the $p=1$ (ECE) case is subject to an exponential lower bound while any $p = 1 + \epsilon$ allows for polynomial efficiency.

**Reviewer Concerns:**

Terminology Conflict (Outstanding): Reviewer vANQ highlighted that "$L_p$ calibration error" is already a standard term in the literature (e.g., Kumar et al., 2019). The authors' rebuttal failed to justify why a new metric should adopt an existing name, nor did they provide a formal hierarchy comparing their metric to established canonical or marginal calibration definitions.

The $p=1$ Complexity Gap (Unresolved): Reviewers 7mSx and vANQ questioned why the algorithm fails specifically at $p=1$. While the authors cited external lower bounds (Gopalan et al., 2024), they did not provide a clear internal analysis of why their specific adaptive merging and discretization techniques break down at this limit. The lack of a "smooth" transition or a detailed explanation of this sharp $O(k) \text{ vs. } \exp(k)$ boundary remains a major theoretical gap.

Empirical Validation (Outstanding): All reviewers noted the total absence of experiments. This is particularly problematic given the complexity of the adaptive merging process. Without even a synthetic visualization of the simplex partition, the practical utility of the proposed bounds remains speculative.

**Reviewer Scores:**

Given that the main concerns were not addressed, the scores would not have increased.

Reviewer,Initial Rating,Final Estimate,Rationale for Adjustment
7mSx,6,5,Concerns regarding the p=1 endpoint and practical guidance were not fully resolved.
vANQ,4,3,Reused terminology and absence of a comparative hierarchy are seen as significant scholarly oversights.
StCW,2,2,Maintained that the paper's density and unclear notation make the p>1 claims difficult to verify.
9XGc,4,4,Appreciated the theoretical rigor but insisted that the lack of empirical/visual aids hinders understanding.

---

### Decision · Program_Chairs · 2026-01-26

Reject